

# Combining ground-based microwave radiometer and the AROME convective scale model through 1DVAR retrievals in complex terrain: an Alpine Valley case study

Pauline Martinet[a], Domenico Cimini[b], Francesco De Angelis[c], Guylaine Canut[a], Vinciane Unger[a], Remi Guillot[a], Diane Tzanos[a], and Alexandre Paci[a]

[a]CNRM UMR 3589, Meteo-France/CNRS, Toulouse, France
[b]IMAA-CNR, Potenza, Italy
[c]CETEMPS, University of L'Aquila, L'Aquila, Italy

*Correspondence to:* Pauline Martinet (pauline.martinet@meteo.fr)

**Abstract.** A RPG-HATPRO ground-based microwave radiometer (MWR) was operated in a deep Alpine valley during the Passy-2015 field campaign. This experiment aims at investigating how stable boundary layers during wintertime conditions drive the accumulation of pollutants. In order to understand the atmospheric processes in the valley, MWR continuously provide vertical profiles of temperature and humidity at a high time frequency, providing valuable information to follow the evolution of the boundary layer. A one-dimensional variational (1DVAR) retrieval technique has been implemented during the field campaign to optimally combine MWR and 1h forecasts from the French convective scale model AROME. Retrievals were compared to radiosonde data launched at least every 3 hours during two intensive observation periods (IOP). An analysis of the AROME forecast errors during the IOPs has shown a large underestimation of the surface cooling during the strongest stable episode. MWR brightness temperatures were monitored against simulations from the radiative transfer model ARTS2 (Atmospheric Radiative Transfer Simulator) and radiosonde launched during the field campaign. Large errors were observed for most transparent channels (i.e., 51-52 GHz) affected by absorption model and calibration uncertainties while a good agreement was found for opaque channels (i.e., 54-58 GHz). Based on this monitoring, a bias correction of raw brightness temperature measurements was applied before the 1DVAR retrievals. 1DVAR retrievals were found to significantly improve the AROME forecasts up to 3 km but mainly below 1 km and to outperform usual statistical regressions above 1 km. With the present implementation, a root-mean-square-error (RMSE) of 1 K through all the atmospheric profile was obtained with values within 0.5 K below 500 m in clear-sky conditions. The use of lower elevation angles (up to 5 °) in the MWR scanning and the bias correction were found to improve the retrievals below 1000 m. MWR retrievals were found to catch very well deep near-surface temperature inversions. Larger errors were observed in cloudy conditions due to difficulty of ground-based MWR to resolve high level inversions that are still challenging. Finally, 1DVAR retrievals were optimized for the analysis of the IOPs by using radiosondes as backgrounds in the 1DVAR algorithm instead of the AROME forecasts. A significant improvement of the retrievals in cloudy conditions and below 1000 m in clear-sky was observed.

From this study, we can conclude that MWR are expected to bring valuable information into NWP models up to 3 km altitude both in clear-sky and cloudy-sky conditions with the maximum improvement found around 500 m. With an accuracy between



0.5 and 1 K in RMSE, our study has also proved MWR to be capable of resolving deep near-surface temperature inversions observed in complex terrain during highly stable boundary layer conditions.

## 1  Introduction

Atmospheric boundary layer (ABL) observations of temperature and humidity profiles at a high temporal resolution are neces-
sary for the improvement of numerical weather prediction (NWP) and for a better understanding of small-scale phenomena. In fact, new generation of convective scale models has been developed in the last ten years in order to improve the forecasts of high impact weather events like heavy convection, precipitation, fog or low clouds. In order to initialize convective scale models through data assimilation algorithms, a denser network of ABL observations is needed as it is the most important under-sampled part of the atmosphere (National Research Council United States (2010)). In parallel, a better understanding of boundary layer
processes is essential to improve parameterisations used to describe the evolution of phenomena at a smaller scale than the model grid. To that end, observations enabling a fine description of the diurnal evolution in the ABL are important to improve our knowledge and understanding of these small-scale phenomena. Among them, ABL processes in mountainous regions are an active area of research due to complex atmospheric dynamics, anabatic and katabatic winds and strong temperature inver-
sions (Rotach and Zardi (2007)). Urban valleys are often affected by severe pollution events during wintertime anticyclonic
conditions while the atmospheric circulation in the valley is decoupled from the synoptic dynamics aloft (Lehner and Gohm (2010), Gohm et al. (2009), De Franceschi and Zardi (2009), Silcox et al. (2012)). This is particularly the case in the Arve River Valley near the city of Chamonix located in the French Alps where the air quality is one of the worst in France. The Passy-2015 field campaign was conducted to improve our knowlegde on how pollutants are accumulated and dispersed during stable episodes in this urbanized valley (Paci et al. (2016)). To better understand and forecast these pollution events, vertical
profiles of temperature at a high temporal resolution can be valuable. In fact, information on the link between the atmospheric stability and the amount of pollutant in the atmosphere can be studied as well as the description of temperature inversions and stratifications (Silcox et al. (2012), Chemel et al. (2016)).
Radiosounding remains one of the most accurate method to measure temperature profiles but their cost and induced finite time resolution (once or twice per day usually for instrumented site) is a limitation for a fine description of the diurnal cycle
of the boundary layer. On the contrary, ground-based microwave radiometers (MWR) can provide continuous observations of temperature and humidity profiles at a high frequency rate (up to 1 s for humidity profiles, a few minutes for tempera-
ture). Even if the vertical resolution decreases with altitude (Cimini et al. (2006)), information from MWR mostly resides in the ABL (Löhnert and Maier (2012)) and atmospheric profiles are provided in both clear and cloudy-sky conditions making them useful for a long-term monitoring of boundary layer dynamics. Atmospheric profiles are generally retrieved from statis-
tical regressions using a long-term database of radiosoundings (Crewell and Lohnert (2007), Löhnert and Maier (2012)). This method relies on a long time serie of radiosonde profiles to represent most of the atmospheric variability. However such a large number of radiosonde profiles is rarely available. NWP models can provide a database of atmospheric profiles when no radiosonde is available (Güldner (2013)). However, this method may not be well suited in complex terrain for which forecast



skills are known to be less accurate particularly due to unrepresented processes associated with subgrid scale orography. One-dimensional variational (1DVAR) retrievals have also been used to retrieve in an optimal way temperature and humidity profiles by combining observations and an *a priori* of the atmospheric state. This *a priori* profile can either be represented by a climatology, radiosounding on instrumented site (Löhnert et al. (2004), Löhnert et al. (2008)) or a short-term forecast from a NWP model. The 1D-Var technique was applied by Hewison (2006), Cimini et al. (2006), Hewison (2007), Cimini et al. (2010) and Cimini et al. (2011) using forecasts from a mesoscale model on various datasets of MWR observations from the MeteoSwiss station of Payerne to observations in Alaska or Vancouver during the 2010 Olympic games. A root-mean-square-error (RMSE) within 1.5 K was obtained for the three experiments by comparison to radiosondes. Recently, Martinet et al. (2015) illustrated for the first time a 1DVAR assimilation of real MWR observations into the convective scale model AROME and obtained a RMSE within 1 K in clear-sky and 1.3 K in cloudy sky up to 6 km, most of the information content brought into the model being located below 3 km altitude.

During the Passy-2015 field campaign, a 14 channel MWR has been operated from December 2014 to March 2015 in a deep and narrow Alpine valley. Although there have already been MWR deployments on complex terrain (Kneifel et al. (2010), Cimini et al. (2011), Massaro et al. (2015)), this study investigates the following questions:

- Can ground-based MWR resolve temperature profiles characterized by sharp temperature inversions during very stable conditions in such a deep and narrow valley, without being affected by surrounding mountains ?

- What added value can MWR bring to NWP models in stable conditions which are known to be a current issue in NWP forecasts ?

The paper begins with an overview of the instrumentation used in the Passy-2015 field campaign (section 2) and the 1D-Var algorithm (section 3) followed by an analysis of the AROME forecast errors during the experiment (section 4). Monitoring of the radiometer brightness temperature measurements enabling the computation of a bias correction is presented in section 5. Finally, performance of 1DVAR retrievals compared to regressions is discussed in section 6.

## 2 Instrumentation

### 2.1 The Passy-2015 field campaign

The Passy-2015 field campaign was designed in order to improve our understanding on how the atmospheric dynamics during wintertime anticyclonic conditions, leading to persistent stable boundary layers, drives the accumulation and dispersion of pollutants in the atmosphere of the Arve Valley around the city of Passy. This French urbanized valley is known for severe pollution episodes with daily concentration of PM10 (aerosols with diameter less than 10 $\mu$m) regularly above 50 $\mu g m^{-3}$. The valley is approximately 2000 m deep and maximum 2 km wide (fig. 1). The ground altitude is approximately 560 m a.g.l down the valley. A large number of instruments were deployed from end of November 2014 to end of March 2015 on five instrumented sites down the valley. Among them, there are microwave radiometers, wind profilers, ceilometer, sodar, lidars,



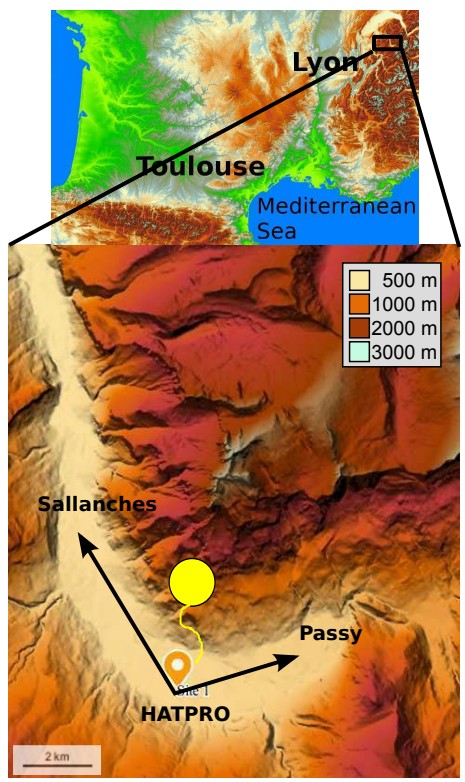

**Figure 1.** View of the area of interest, close to the city of Passy in the Arve river valley. Microwave radiometer and radiosondes were deployed on measurement site 1. Topographic maps from www.geoportail.gouv.fr, IGN 2017

tethered ballons, instrumented towers. A detailed presentation of the field campaign can be found in Paci et al. (2016). Two intensive observation periods (IOP) have been carried out during the campaign. The observing system was reinforced during these periods with radiosondes launched every 3 hours and up to 1.5 hours. The first IOP took place from the 6th to the 14th of February, the second one from the 17th to the 20th of February.

## 2.2 HATPRO MWR

A HATPRO MWR (Rose et al. (2005)) was deployed on site 1 (Fig. 1) and is oriented to scan the Passy valley in two opposite directions: Passy in the NorthEast and Sallanches in the NorthWest direction. The HATPRO MWR measures downwelling brightness temperatures in 14 channels. The first seven are located on the upper-frequency wing of the 22.24 GHz water vapor absorption line (called K-band), the last seven at the 60 GHz oxygen complex band (called V-band). K-band channels are used to retrieve atmospheric humidity and liquid water content while V-band channels are used for atmospheric temperature retrievals. Observations are made either in zenith mode pointing at 90 ° or in boundary layer mode scanning the atmosphere





under lower elevation angles from 90 ° to 5.4 °. One boundary layer scan is performed in each direction approximately every 10 minutes. The use of boundary layer scan was found to significantly improve the accuracy of temperature profiles in the first km assuming that the atmosphere is horizontally homogeneous around the MWR (Crewell and Lohnert (2007)). Even if this assumption is not necessarily valid in complex terrain, the study of Massaro et al. (2015) has shown a good accuracy of

temperature profiles with no degradation due to the nearby mountain. The radiometer needs to be well calibrated to exploit the optimal calibration coefficients in order to convert detected intensities into brightness temperatures. To that end a liquid nitrogen cooled load considered as a blackbody at the boiling temperature of 77 K is generally used (Küchler et al. (2016)). A liquid nitrogen calibration was performed at the beginning of the experimental campaign at the end of November 2014.

### 2.3 Ancillary Data

In addition to the HATPRO MWR, observations by 84 radiosonde ascents are used to validate temperature profiles retrieved by the MWR. VAISALA RS92 radiosondes with an expected accuracy of 0.5 K in temperature and 5 % in relative humidity were launched approximately every 3 hours and up to 1.5 hours during the IOPs. Radiosondes were launched approximately at 00, 03, 06, 09, 12, 15, 18 and 21 UTC. They provide vertical profiles of pressure, temperature, relative humidity, and wind profiles at approximately 10 m vertical resolution. The temperature at 1.5 m is provided by an external weather station combined with

the RS measurements through the VAISALA software. A new system to increase the frequency of radiosondes by recovering previously launched probes has been used during the field campaign (Legain et al. (2013)). In order to be able to pick up the probes, they should not drift too far away from the launching site. As a consequence, most of the radiosondes were released at about 2 km altitude, to make sure they can be picked up in the valley.

A ceilometer Vaisala CT25K deployed a few meters from the MWR is also used to determine the cloud base altitude. This cloud

base can be used to optimize the 1DVAR retrievals in cloudy conditions and to separate clear-sky from cloudy-sky observations when analysing the results.

## 3 Retrieval Algorithm

### 3.1 1DVAR framework

A comparison of several methods to convert brightness temperatures into temperature and humidity profiles have proved the

1DVAR technique to be the optimal one (Cimini et al. (2006), Martinet et al. (2015)) when the *a priori* profile and uncertainty estimates are suitable. The 1DVAR framework used in this study is based on the optimal estimation theory by Rodgers (2000). MWR observations are combined with an *a priori* estimation of the atmospheric state which can be either a short-term-forecast or a previous radiosonde profile. In this context, *a priori* refers to the first guess of the iterative algorithm representing a good estimate of the atmospheric conditions as the starting point of the minimization. Each source of information is weighted by

corresponding uncertainty called the background-error-covariance matrix (**B**) for the *a priori* profile and the observation-error-covariance matrix (**R**) for the observation to find the optimal state. The background-error-covariance matrix represents the





auto-covariances and cross-covariances of the first guess errors. Thus, it defines the variances of the first guess errors at each vertical level for each variable, the vertical correlations of the first guess errors at different levels and the correlation of these errors between different variables (temperature and humidity for example). An observation operator including interpolations from model space to observation space and a radiative transfer model is needed to compute the equivalent observation from

the *a priori*. The method iteratively modifies the state vector $x$ from the *a priori* $x_b$ to minimize the following cost function:

$$J(\mathbf{x}) = \frac{1}{2}(\mathbf{x} - \mathbf{x}_b)^T \mathbf{B}^{-1}(\mathbf{x} - \mathbf{x}_b) + \frac{1}{2}(\mathbf{y} - \mathrm{H}(\mathbf{x}))^T \mathbf{R}^{-1}(\mathbf{y} - \mathrm{H}(\mathbf{x}))$$

where H represents the observation operator, $^T$ represents the transpose operator and $^{-1}$ the inverse operator. The observation-error-covariance matrix $\mathbf{R}$ should take into account representativeness and forward model errors as well as radiometric noise.

During the minimisation process, a Levenberg-Marquardt descent algorithm is applied by introducing a factor $\gamma$ that is

adjusted after each iteration. If the cost function is not decreased with the new profile, the factor $\gamma$ is multiplied by 10. The iterative solution that minimizes the cost function $J$ is given by:

$$
\begin{aligned}
\mathbf{x}_{i+1} \quad = \quad & \mathbf{x}_i + \left((1+\gamma)\mathbf{B}^{-1} + \mathbf{H}_i{}^T \mathbf{R}^{-1} \mathbf{H}_i\right)^{-1} \times \\
& \left(\mathbf{H}_i{}^T \mathbf{R}^{-1}(\mathbf{y} - \mathrm{H}(\mathbf{x}_i)) - \mathbf{B}^{-1}(\mathbf{x}_i - \mathbf{x}_b)\right)
\end{aligned}
\tag{1}
$$

where $\mathbf{H}_i$ is the Jacobian matrix which represents the sensitivity of the observation operator to changes in the control vector $x$

($\mathbf{H}_i = \partial \mathrm{H}(x_i)/\partial x_i$).

## 3.2 NWP model

In this study 1-hour forecasts from the French convective scale model AROME (Application of Research to Operations at MEsoscale, Seity et al. (2011)) are used as *a priori* profiles or "backgrounds". AROME is a limited area model covering Western Europe with non-hydrostatic dynamical core. Since beginning 2015, the horizontal resolution of AROME has been

increased from 2.5 km to 1.3 km as well as the number of vertical levels from 60 to 90 (Brousseau et al. (2016)). This increase in horizontal and vertical resolutions is particularly useful to better represent complex terrains. Vertical levels follow the terrain in the lowest layers and isobars in the upper atmosphere. The detailed physics of Arome are inherited from the research Meso-NH model (Lafore et al. (1997)). Deep convection is assumed to be resolved explicitly, but shallow convection is parameterized following Pergaud et al. (2009). A bulk one-moment microphysical scheme (Pinty and Jabouille (1998))

governs the equations of the specific contents of six water species (humidity, cloud liquid water, precipitating liquid water, pristine ice, snow, and graupel). This new version also performs 3D-Var analyses every hour instead of every three hours to optimize the use of frequent observations. All conventional observations are assimilated together with wind profilers, winds from space-borne measurements (Atmospheric Motion Vectors and scatterometers), Doppler winds (Montmerle and Faccani (2009)) and reflectivity (Wattrelot et al. (2014)) from ground-based weather radars, satellite radiances as well as ground-based


GPS measurements (Mahfouf et al. (2015)).

### 3.3 Settings

In this study the control vector $x$ consists in temperature and humidity profiles on the same 90 levels as defined in AROME. These levels cover the atmospheric range from the ground up to 30 km, the vertical resolution decreasing with altitude: 20-100 m below 1 km, 100-200 m from 1 to 5 km, around 400 m at 10 km. The observation vector $y$ consists in brigthness temperatures (BT) in all V-band channels (51.26, 52.28, 53.86, 54.94, 56.66, 57.3, 58 GHz) at zenith and only opaque channels (above 54 GHz) at low elevation angles: 42, 30, 19.2, 10.2 and 5.4 °. This study only focusses on temperature profiles, thus only V-band channels are used. The forward model operator used in this study is the line-by-line Atmospheric Radiative Transfer Simulator 2 (ARTS, Eriksson et al. (2011)) and 1DVAR experiments are performed using the Qpack2 package (Eriksson et al. (2005)) provided with the ARTS software. For the radiative transfer simulations, the gaseous absorption is calculated according to Rosenkranz (1998) for $O_2$ and water vapour. In simulations taking into account the liquid water absorption, the model of Liebe et al. (1993) is used.

The observation-error covariance matrix $\mathbf{R}$ is assumed to be uncorrelated with a standard deviation of 0.5 K for channels 8 to 9 and 0.2 K for channels 10 to 14. These values have been chosen empirically on the basis of previous studies by Löhnert et al. (2008) and Hewison (2007). The same values have been used in Martinet et al. (2015) with the instrument used in this study and have shown to be good estimates of the observation errors. In the future, a dedicated study will be performed to review these values and quantify the correlations in noise between the different channels by continuously measuring the BTs of the internal black body target.

### 4 Evaluation of the AROME model during the Passy-2015 field campaign

In real-time during the Passy-2015 field campaign, temperature profiles were retrieved from the MWR measurements using linear regressions implemented within the HATPRO proprietary software. The regression coefficients were provided by the RPG manufacturer to the national service MeteoSwiss and are based on the 1989-2005 Payerne radiosonde data via radiative transfer calculations. The Payerne coefficients were chosen due to the lack of radiosonde data close to the city of Passy and for the similar climatic conditions between Passy and Payerne.

In order to evaluate the performance of the AROME model during the Passy-2015 experiment figure 2 shows the time series of temperature profiles observed by radiosondes, retrieved from the HATPRO MWR by the Payerne linear regression coefficients and extracted from the AROME analyses during the first IOP. The stable episode starts the 9th of February and ends the 13th of February. During this event a persistent inversion is observed, however we can note that stability is depleted in the first 500 meters every day between noon and 3 to 5 pm due to the solar heating. The diurnal cycle and a very cold air mass (up to -10 ° C) close to the surface at night are very well detected by the MWR. We observe a good agreement of the overall atmospheric structure between radiosonde data and MWR observations. The root-mean-square differences (RMSE) between





the regressions and the radiosondes are 0.7 K below 500 m except the first two points close to the surface, below 1.3 K at 1200 m and increase up to 2 K at 4000 m. These values are consistent with those reported in Löhnert and Maier (2012) from another HATPRO radiometer operated in a less complex terrain and from Massaro et al. (2015) in a truly complex terrain in the Inn Valley. This result confirms that microwave radiation that could originate from nearby slopes does not seem to degrade the

quality of MWR inversions. MWR can thus be safely deployed in complex terrain and similar temperature accuracy to that of flat and less complex terrain can be expected.

Figure 2 also demonstrates that the 10 minute resolution of the MWR observations during the field campaign is a real advantage to complete the radiosonde time serie for a detailed description of the boundary layer diurnal cycle. During IOP 1, the 2015 operational version of the AROME model missed the large cooling of the surface at nighttimes. The AROME model

demonstrated difficulties in properly representing such conditions which is a well known issue of current NWP models. It induces large differences between the radiosonde observations and the AROME forecasts by up to -12 K at the surface during the strongest stable event (10th, 11th and 12th of February).

To quantify the accuracy of the AROME analyses in the valley during IOP1 figure 3 shows temperature differences between radiosonde and AROME at three different levels: 1.5 m a.g.l., 1000 m a.g.l. and 1500 m a.g.l. The measurement at 1.5 m comes

from an external weather station well ventilated. To interpret these temperature errors, the differences between the radiosonde temperature measurement at the boundary layer height $zi$ and the surface measurement from the external weather station: $\Delta T = T_{RS}(zi) - T_{station}(1.5m)$. To estimate during the day the thin convective layer top which develops under the effect of solar heating, we used one of the standard definitions given by Stull (2012) and Sullivan et al. (1998) as the height of the maximum gradient of potential temperature. The estimation of the boundary layer height in stable conditions is more tricky and

has been a longstanding problem with definitions varying according to the application. Here, the stable boundary layer top has been defined as the top of the surface inversion of the stable layer using the definition from Beyrich (1997). This definition is used when a positive temperature gradient near the surface is found. The temperature difference $\Delta T$ quantifies the atmospheric boundary layer stability. Negative values indicate convective conditions while positive values indicate stable conditions more pronounced when the temperature difference is larger. The term "stability index" will be used in our analysis. We can note that

the surface error is relatively low at the beginning of the period (smaller than 5 K) and increases with the atmospheric stability. The stability index increases from the 9th to the 11th of February and decreases after to reach values similar to the beginning of the episode. The stability index changes from positive (i.e. stable) to negative (i.e. unstable) every day between 12 and 18 UTC approximately. The temperature errors of the AROME analyses at the surface are consistent with the evolution of the atmospheric stability. The largest errors reach -12 K the 11th of February at 03 UTC (Fig. 4) when the stability is maximal

with a value of 14 K.

On the contrary, the evolution of the 1000 m temperature error is not correlated with the atmospheric stability with larger errors during unstable conditions before the 9th of February. At the beginning of the IOP cloudy conditions with low level clouds located around 1000 m were observed. It results in a sharp temperature inversion at the cloud base (Fig. 4) which is also a known source of error in NWP forecasts. At 1500 m a.g.l, the error stays within 2 K during all the period showing a good

accuray of the AROME analyses at an altitude corresponding roughly to the averaged valley crest.



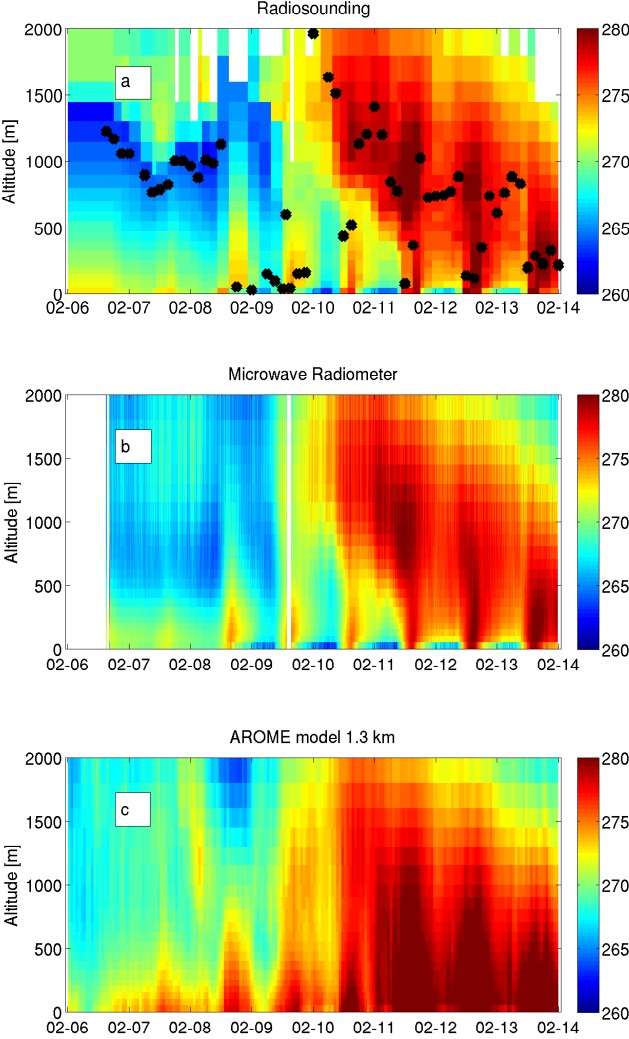

**Figure 2.** Time series of temperature profiles during IOP 1: (a) from radiosounding with corresponding boundary layer heights (black crosses), (b) from microwave radiometer, (c) from AROME analyses. Altitudes are given in m above ground level.

To summarize, the accuracy of the AROME analyses is degraded inside the valley which is affected by an atmospheric circulation decoupled from the synoptic dynamics above the valley crest. The degradation of the AROME analyses is correlated with the establishment of the stable episode. The surface cooling is strongly underestimated by AROME in this context. However, above the top of the valley, the analysis errors are much smaller and correspond to the expected accuracy of the model. This result confirms the fact that MWR can bring valuable information in the altitude range where the NWP error is the largest and where a lack of observations is still observed in operational networks.




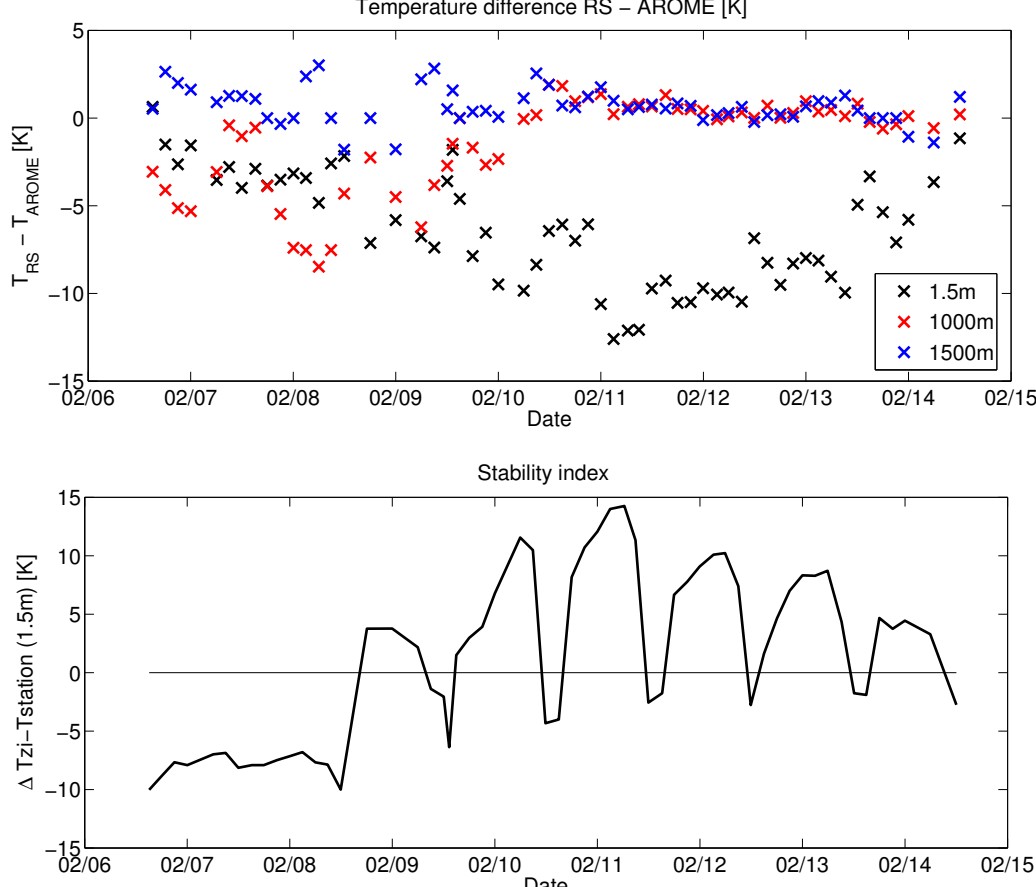

**Figure 3.** Top panel: Temperature differences between radiosondes and AROME analyses at three levels: 1.5 m in black, 1000 m in red, 1500 m in blue. Bottom panel: temperature differences between the radiosonde measurement at the boundary layer height and the surface measurement from an external weather station. Altitudes are given in m above ground level.

## 5 Observation minus background monitoring

### 5.1 Data screening

In order to remove discrepancies in the forward simulations due to cloud mislocations in the forecast model, a screening of MWR observations between clear and cloudy-sky cases has been performed. First of all, a sanity check is performed to remove
5 MWR observations for which the rain flag provided within the instrument datastream was activated. As the HATPRO configuration was optimized to retrieve temperature profiles at a high vertical resolution, few zenithal observations were performed between two boundary layer scans. The small amount of data at zenith does not allow the use of the standard deviation of MWR BT measurements at 31 GHz to detect possible clouds in the field of view of the instruments (Ebell et al. (2017)). The



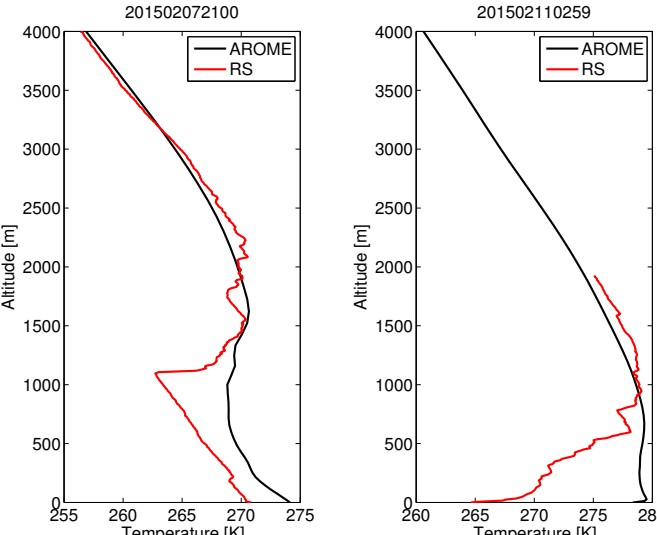

**Figure 4.** Temperature profiles observed by the radiosonde (red line) or extracted from the AROME analyses (black line): during unstable conditions the 07th of February at 21 UTC (left panel) and during the most stable periode the 11th of February at 03 UTC (right panel). Altitudes are given in m above ground level.

cloud base height provided by the CT25K ceilometer was thus used as a reference to identify cloudy-sky observations. If the lowest cloud base height during a +/- 20 min window around the MWR observation is smaller than 6000 m, the observation is classified as cloudy. In case no ceilometer observation is available, the infrared radiometer temperature provided with the HAT-PRO platorm has to be smaller than -30 ° C to consider the observation as liquid free (similar approach used in Martinet et al.

(2015)).The result of this classification of radiosondes between clear-sky and cloudy-sky observations is shown in figure 5 in addition to the ceilometer cloud base height. Among the 84 radiosondes launched during the Passy-2015 field campaign, 56 were classified as clear-sky.

## 5.2   O-B analysis from AROME forecasts

Monitoring observation minus background departures is an important step before any assimilation. First of all, the best esti-
mate of the analysis state is obtained only if background and observation errors follow Gaussian distribution with zero mean. Quality-controlled and bias-free observations are thus necessary to obtain good estimate of atmospheric profiles. Should not this be the case, a bias correction of the observations can be proposed to meet the requirements of variational assimilation. While Löhnert and Maier (2012) and Navas-Guzmán et al. (2016) used radiosonde to simulate the equivalent brightness temperature spectrum, Martinet et al. (2015) showed the possibility of using the AROME forecasts instead of radiosonde data.
Using AROME forecasts enables the detection of BT bias offset when no radiosonde is available close to the MWR site. However, a new source of error is added coming from possible systematic NWP errors. Even though differentiating the different





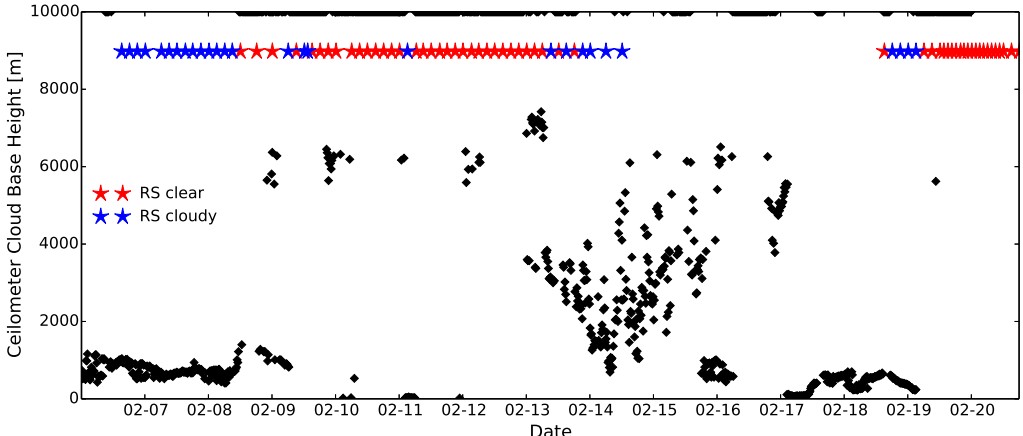

**Figure 5.** Cloud base height retrieved from ceilometer CT25K during the two IOPs of the Passy-2015 experiment. Stars represent the launch times of the radiosondes during the campaign classified as clear-sky in red and cloud-sky in blue. When no cloud is found by the ceilometer (clear-sky), a value of 10 000 m has been chosen by default.

sources of errors ( instrumental, forward model and background errors) can be complex, this monitoring is widely used in the satellite data community.

BT simulations were performed with the ARTS radiative transfer model and 1-hour AROME forecasts (temperature, humidity) using 2 months of data (February and March 2015). The closest AROME grid point in the valley with an altitude difference of

only 2 m compared to the MWR location was used. Figure 6 shows the observation minus background departures (O-B) as a function of the atmospheric boundary layer stability for one transparent channel (51.26 GHz) and one opaque channel (58 GHz) and different elevation angles. Only clear-sky observations are considered with a screening procedure described in section 5.1. As radiosondes are not available throughout the period, the atmospheric stability is computed from MWR temperature profiles retrieved by linear regression. The temperature difference between 500 m and 50 m is used. Different altitudes have been tested

but 500 m was found to describe best the development and destruction of stability in the boundary layer at least during the IOPs. The MWR temperature retrieval at surface was not used as large errors have been observed in Martinet et al. (2015) and would impact the evaluation of the stability. Instead, the second level of the MWR retrievals (50 m) has been chosen as it has shown a better accuracy with respect to radiosonde measurements. From this figure, we can observe that the O-B departures at 58 GHz are highly correlated to the atmospheric stability which is not the case at 51.26 GHz. As opaque channels are more

sensitive to the lowest atmospheric layers, this result indicates that the forward simulations are highly affected by the larger AROME forecast errors in the boundary layer during stable episodes. On the contrary, the accuracy of the AROME forecasts in the upper layers stay stable during the period. The forward simulations at 51.26 GHz are thus quite stable during the whole campaign. Larger errors are also found with decreasing elevation angles for both transparent and opaque channels. For opaque channels, this can be explained by an increased sensitivity to atmospheric layers close to the surface where the largest errors in





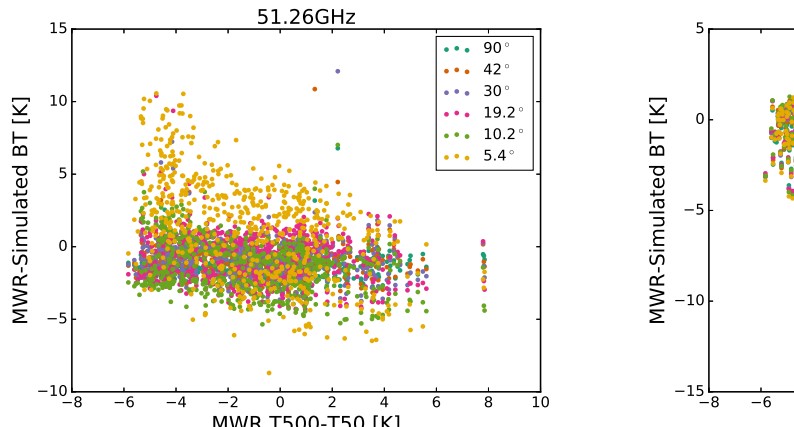
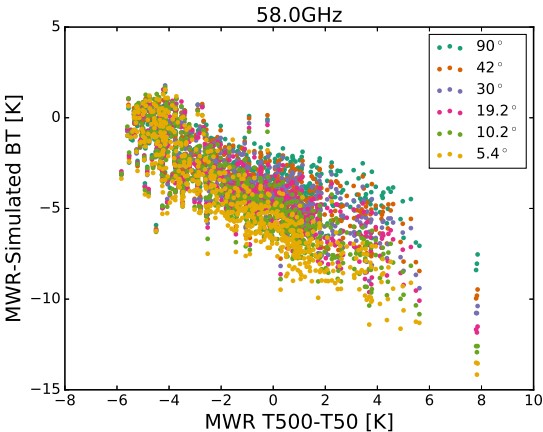

**Figure 6.** Observation minus background (AROME forecasts) departures as a function of HATPRO temperature differences between 500 m and 50 m and different elevation angles from 90 ° to 5.4 °.

the AROME forecasts are observed. For transparent channels, radiations from surrounding slopes can degrade the observations and atmospheric inhomogeneities can cause larger discrepancies with the simulation.

This section has shown that, in the particular case of the Passy-2015 experiment, the use of AROME forecasts to infer any systematic BT offset is not appropriate. In fact, the large forecast errors during wintertime stable episodes exceed the instrumental
errors. The computation of O-B departures on a larger time period could probably smooth the forecast errors to only highlight instrumental errors. In order to correctly infer any BT offset, the O-B departures are computed from the radiosondes launched during the IOPs in the next section.

### 5.3  O-B analysis from radiosondes

Observation minus background departures have been computed from radiosonde profiles launched during the field campaign.
As most of the radiosondes did not reach more than 2 km altitude above ground, it is important to complete the atmospheric profiles up to 30 km to avoid large discrepancies in the simulation of transparent channels. Radiosondes were interpolated into the AROME vertical grid below 2 km and completed with AROME analyses above. As the accuracy of AROME analyses is better than 1 K above 2 km, this combination should not degrade the forward simulations. Figure 7 shows the bias and standard deviation of O-B departures at different elevation angles. We can note a stronger dependency to the elevation angles
for transparent channels (51 to 52 GHz). These channels are more sensitive to calibration errors for decreasing optical depth (higher elevation angles). The largest bias (-4.2 K) is found at 52.25 GHz and an elevation angle of 90 ° while it is below 0.5 K for opaque channels. Standard deviations within 1 K are observed for all channels and all elevation angles except at 51.25 GHz for elevation angles lower than 19.2 ° and at 52.25 GHz at 5.4 °. This degradation can be due to an increase in atmospheric inhomogeneities explaining that transparent channels are generally not used at low elevation angles (Crewell and Lohnert





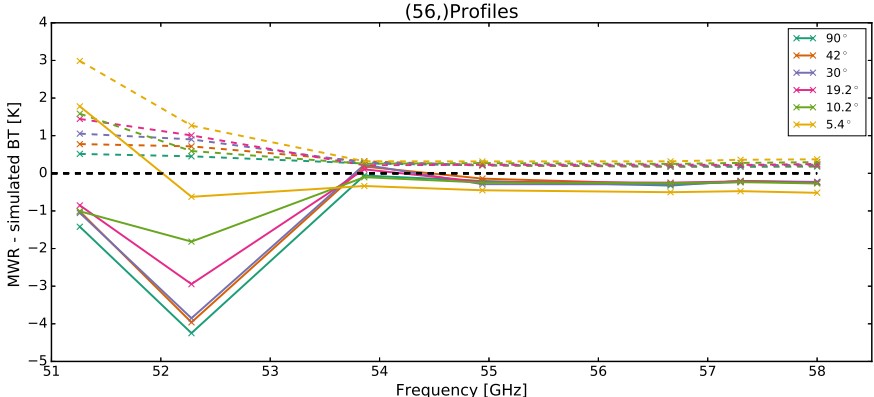

**Figure 7.** Observation minus background departures from radiosondes combined with AROME analyses for different elevation angles from 90 ° to 5.4 ° (coloured lines). For each angle, the bias is shown in solid line and the standard deviation in dashed line.

(2007)). For opaque channels, bias and standard deviation smaller than 0.2 K are observed at all angles except at 5.4 ° where the bias reaches 0.5 K. Similar values were found in the study of Martinet et al. (2015) with the same HATPRO instrument on a less complex terrain. The consistency between both studies points out a good stability of the instrument despite several deployments and calibrations. An improvement in the calibration procedure has also been observed with a significant decrease

of standard deviation for all channels (up to 3 times at 90 °). A similar bias shape was found in Löhnert and Maier (2012) and Navas-Guzmán et al. (2016) with a large negative bias at 52.25 GHz but also on several sites in Europe (De Angelis et al. (2017)). This large bias can be due to a combination of calibration errors and absorption model uncertainties (Hewison 2006). This analysis demonstrates that a constant bias correction can be safely applied to the set of measurements used for temperature retrievals: only zenith angle for frequency below 53 GHz, and all elevation angles above. It will be applied and discussed in

the next sections.

## 6  1DVAR retrievals

### 6.1  Background errors

In the operational AROME model, the background-error-covariance matrix **B** is computed from an ensemble assimilation that considers explicit observation perturbations and implicit background perturbations through the cycling (Brousseau et al.

(2011)). The AROME ensemble assimilation is coupled to the operational ensemble assimilation at global scale AEARP (Berre et al. (2007)). However, the expected background accuracy (diagonal terms of the **B** matrix) suggests a forecast error of less than 1 K in the boundary layer on average through all the AROME domain. This operational **B** matrix significantly underestimates the AROME forecast errors during the Passy-2015 experiment. A new **B** matrix has thus been computed from the differences between the AROME forecasts and the radiosonde data similarly to Cimini et al. (2011). The bias and standard





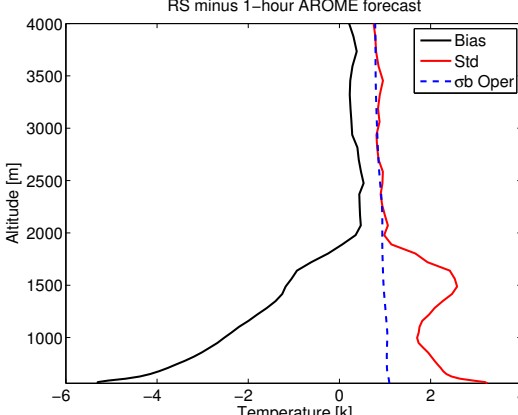

**Figure 8.** Bias (black line), standard deviation (red line) of the radiosonde minus 1h-AROME temperature differences. The operational background error used in AROME for the assimilation of satellite data is shown as the blue dashed line.

deviation of these differences compared to the operational background errors used for the assimilation of satellite data are shown in figure 8. In temperature, a large bias of approximately -5 K is observed at the surface and corresponds to the large overestimation of the temperature by AROME during the stable episodes. A standard deviation of 2 to 3 K, which is two to three times larger than the expected background error, is observed between the surface and 1700 meters. The temperature error at higher altitude is much smaller ($\sim$ 1 K) and closer to the value prescribed in the operational assimilation system, corresponding to a decrease in the forecast error above the valley crest. Similar features were found with ECMWF and NCEP models in an Arctic environment in the study of Cimini et al. (2010). As the 1DVAR retrieval accuracy depends on how well the **B** matrix is defined, the diagonal terms of the **B** matrix (auto-covariance of the temperature errors) were modified below 2 km altitude with the standard deviation of the radiosonde minus AROME differences. In order to provide statistically consistent increments at the neighbouring levels of the model, the vertical correlations of the operational **B** matrix were conserved.

## 6.2 Sensitivity of retrievals to elevation angles and bias correction

The 1DVAR method with the settings previously described has been applied to MWR observations during the two IOPs of the Passy-2015 campaign. The retrievals are evaluated against radiosondes and compared to the linear regressions with the Payerne coefficients. According to section 5, a constant bias correction is removed from all channels before the 1DVAR algorithm. This bias correction is not used for the linear regressions that directly come from the raw measurements and the HATPRO proprietary software. Figure 9 evaluates temperature retrievals against radiosondes in terms of bias and RMSE focussing only on clear-sky profiles. 1DVAR retrievals are compared to AROME 1h forecasts used as backgrounds in the algorithm and to linear regressions. To evaluate the impact of the bias correction, 1DVAR retrievals with and without bias correction are also compared while the impact of low elevation angles is investigated by comparing retrievals using only zenith angle or all angles.





As previously mentioned, we observe a positive bias up to 6 K in the AROME backgrounds decreasing with altitude to reach -0.5 K above 1200 m. Very similar values of mean deviations are found between both 1DVAR retrievals when a bias correction is applied to the measurements despite a decrease of the bias in the first 100 m with additional low elevation angles. Regressions show also almost identical values demonstrating a better behaviour compared to the large bias observed with neural networks

in Martinet et al. (2015). Without applying a bias correction to the measurements, a small degradation in the bias is observed below 1000 m. The maximum degradation is found at 1000 m where the bias reaches 0.7 K instead of -0.1 K. The RMSE profiles of 1DVAR retrievals indicate that a significant amount of information can be extracted from MWR observations to improve the AROME backgrounds even though large errors below 1500 m are observed (from 1 to 8 K) in the *a priori* profile. This situation is quite extreme as the background is very far from the truth at the beginning of the minimisation. However, the

largest background errors are found in the 0-1000 m range where MWR can constrain the most the minimisation due to the high information content of the instrument in this altitude range. Above 1000 m 1DVAR outperforms regressions whose RMSE values increase up to 2.5 K at 6000 m. The performances of 1DVAR retrievals are similar when using the standard deviations from the operational **B** matrix and a custom **B** matrix computed comparing AROME and RS profiles (not shown). This suggests that the operational **B** matrix may be safely adopted for other sites where RS profiles are not available for computing a custom

**B** matrix. However, in the future, it would be interesting to investigate the sensitivity of the 1DVAR retrievals to the flow dependency of the **B** matrix and particularly to the vertical correlation lengths.

For the sake of clarity figure 10 shows MWR retrievals either with regression or 1DVAR with a focus on the range 0 - 1000 m. 1DVAR RMSE values are smaller than 0.8 K below 500 m and within 1 K through all the atmospheric profile. Large RMSE values are found close to the surface for all retrieval methods (up to 1.6 K). The best accuracy of 1DVAR retrievals is found

when the bias correction is applied to the measurements and using all elevation angles up to 5.4 °. A degradation below 1000 m in the 1DVAR retrievals is observed when only observations at zenith are used in the minimisation. In this case, the RMSE values can reach 1.2 K instead of 0.8 K with low elevation angles. This result demonstrates the benefit of low elevation angles to resolve temperature inversions below 1000 m. Below 1000 m, regressions perform slightly better (differences between 0.1 and 0.2 K in RMSE) than 1DVAR. Overall, 1DVAR retrievals provide the best estimate of the atmosphere.

In order to investigate more in details the large RMSE values observed close to the surface, time series of the temperature difference between the surface (weather station) and the first HATPRO level are investigated in figure 11. This difference is compared to the differences between the surface station and tower measurements at 2.5 and 5 m. The evolution of the temperature error is compared to the diurnal cycle of the temperature difference between the surface (1.5 m) and the RS measurement at the first level above ($\sim$ 10 m a.g.l). Positive values indicate stable atmosphere while negative values indicate

convective conditions. We can note a correlation between the decrease in the stability and the increase in the HATPRO surface error. Maximum differences (- 9 K) are found when the stability is minimum corresponding to a maximum of convective activity during daytime. The MWR seems to significantly underestimate the surface warming during the transition phase from stable to convective conditions. However, MWR retrievals can easily be combined with surface sensors providing an higher accuracy at the surface. Figure 11 also shows the differences of the tower measurements at 2.5 and 5 m with respect to the

surface station. The standard deviation is slightly larger with the measurement at 2.5 m compared to the 5 m measurement



**Figure 9.** Vertical profiles of bias (left panel) and root-mean-square-errors (right panel) of the AROME background (red line) and MWR retrievals using all elevation angles against radiosondes: 1DVAR retrievals from AROME 1h forecasts and bias correction (magenta), without bias correction (cyan), with bias correction but using only zenith angle (blue) and linear regressions (black). Results on 56 clear-sky temperature profiles.

(1.1 K instead of 0.9 K respectively). To test the feasibility of combining surface measurements and MWR observations in a physical way, the tower measurement at 5 m was included in the observation vector with a sharp surface-peaked Jacobian associated. The 1DVAR retrievals look very similar to what was previously shown but a significant improvement in the RMSE at the surface was found with a decrease from 1.6 K to 1 K as expected (this configuration is used later on in figure 15). In the future, this combination could thus be used by deploying a well calibrated surface station in parallel to the MWR.




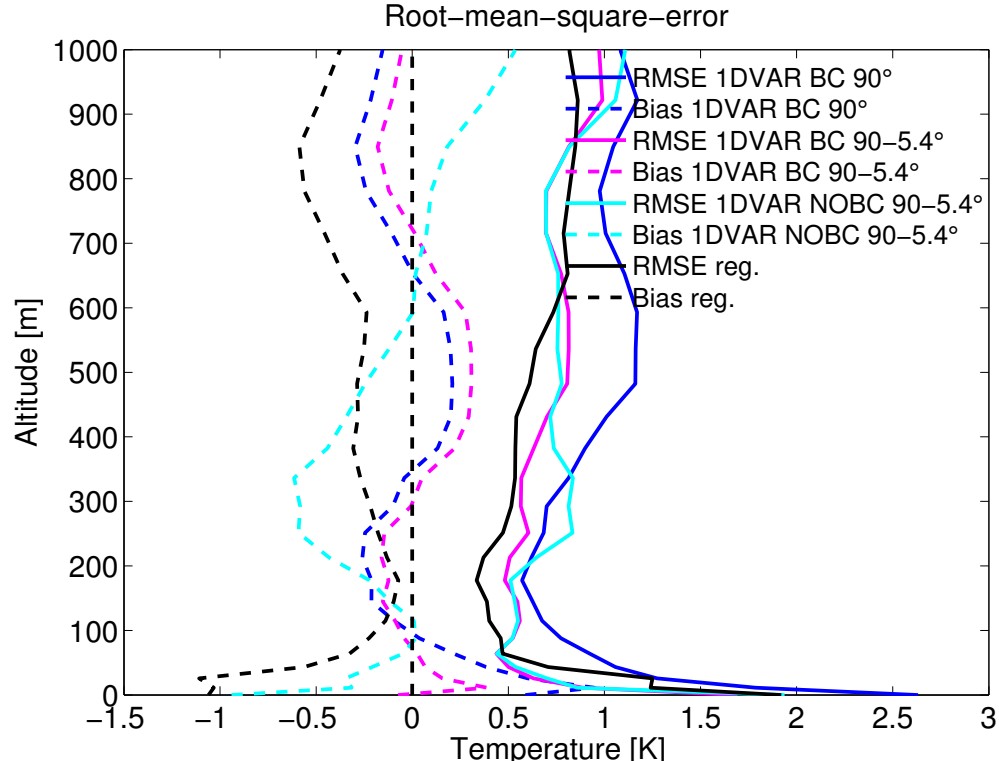

**Figure 10.** Vertical profiles of bias (dashed lines) and root-mean-square-errors (solid lines) of MWR retrievals against radiosondes with a focus on the range 0-1000 m. 1DVAR retrievals from AROME 1h forecasts and bias correction (magenta), without bias correction (cyan), with bias correction but using only zenith angle (blue) and linear regressions (black). Results on 56 clear-sky temperature profiles.

## 6.3 Sensitivity of retrievals to the *a priori*

The previous section has investigated the capability of MWR observations to be assimilated into NWP models by following a similar approach to operational assimilation systems (3DVAR, 4DVAR). However, in the context of field campaigns and the study of boundary layer processes, it can be interesting to get the best possible 1DVAR retrievals by using a more appropriate background profile. In the case of the Passy-2015 field campaign, thanks to the high temporal resolution of RS, the previously launched radiosonde can be used as the background profile instead of the AROME 1h forecast to start the minimisation from a more reasonable *a priori* profile. The **B** matrix has also been recomputed according to the differences between two successive radiosondes in order to be consistent. In order to evaluate this new configuration figure 12 compares the accuracy of 1DVAR retrievals if either 1h AROME forecasts or the previously launched radiosondes are used as backgrounds. When radiosondes are used as backgrounds, the bias is decreased during the analysis providing the best accuracy compared to the other retrievals. In terms of RMSE, the 1DVAR accuracy is improved between 400 m and 1200 m and outperforms the regressions through all the atmospheric profile except a slight degradation at 1200 m. Using RS as backgrounds, RMSE values are below 0.6 K in the





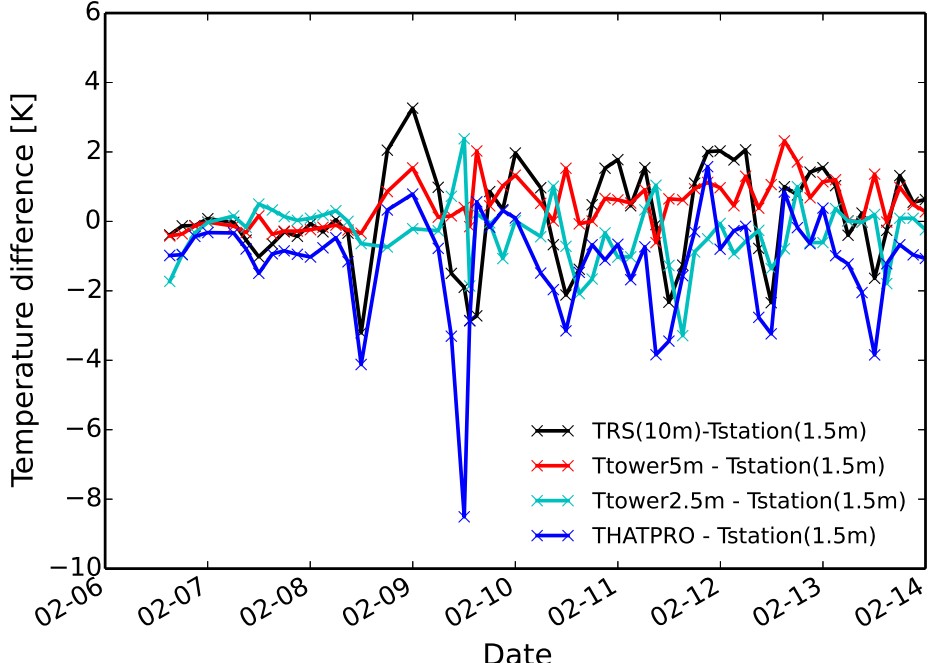

**Figure 11.** Time series of temperature differences between the first HATPRO level and a surface weather station (blue line), between tower measurements at 2.5 and 5 m and the surface station (cyan and red lines respectively). Radiosonde temperature difference (black line) between the surface and the first level above.

first 1000 m and within 1 K above.

An attempt to use the 1DVAR algorithm in cloudy conditions is shown in figure 13. The liquid water path is estimated from HATPRO with a simple and classical dual channel algorithm using brightness temperature measurement at 23 and 31 GHz (Westwater (1978)). The liquid water content profile is estimated from the background temperature and humidity profiles from

5  a modified adiabatic assumption (Karstens et al. (1994)) in layer where the relative humidity exceeds 95 %. The computed liquid water content profile is then scaled with the estimated liquid water path. The obtained liquid water content profile is translated vertically to fit the cloud base height provided by the ceilometer. In case no atmospheric layer exceeds the 95 % relative humidity threshold, a cloud layer is placed at the cloud base height provided by the ceilometer with a geometrical thickness of one layer and a liquid water path equal to the one derived from HATPRO measurements. As MWR are mostly

10  sensitive to the integrated liquid water content rather than the vertical distribution of clouds, this configuration should be sufficient to take into account the cloud contribution in the radiative transfer. Note that only 25 profiles are taken into account in the statistics which makes the dataset too small for a good representativness. A large degradation of the 1DVAR retrievals





from AROME forecasts is observed in cloudy-conditions with an increase of RMSE values up to 3 K at 1000 m. Contrary to what one may think, this degradation does not seem to be directly related to large background errors in the liquid water content but more to the misrepresentation of cloud-based temperature inversions in AROME. In fact, figure 5 shows that most of the cloudy profiles are located at the beginning of the first IOP between 07 and 09 February. This period corresponds to

the strong temperature inversion at 1000 m altitude missed by AROME (fig. 4) inducing large forecast errors up to - 9 K. As the information content from the microwave radiometer is maximum below 1000 m and decreases with altitude, the MWR likely does not bring enough information during the analysis to correct the background profile. In addition to this decrease in information content, the large RMSE value at 1000 m is likely due to the smoothing error related to the low vertical resolution of MWR. This is evident in the bias, showing large positive to negative values going from 500 to 1500 m altitude.

To confirm that this degradation does not come from large errors in the liquid water content background profile, 1DVAR retrievals have been performed using only opaque channels (54-58 GHz). These channels are known to be less affected by cloud-liquid water emission contrary to transparent channels. Figure 14 shows the RMSE of 1DVAR retrievals with this reduced channel set differentiating the results between clear-sky and cloudy-sky conditions. In clear-sky conditions, using only opaque channels, a slight improvement between 900 and 1600 m and a degradation between 1600 and 3000 m are observed although

these differences stay small (0.1 K in RMSE). In cloudy-sky conditions, the same 1DVAR statistics are found with the different channel configurations. As transparent channels are more affected by cloud liquid water emission, we could have expected to observe a larger degradation when these channels are used if the liquid water content is not well modelled. As few differences are observed with transparent channels included in cloudy-conditions, it supports that the degradation in cloudy-conditions is likely to come from sharp elevated temperature inversions.

In cloudy-sky, regressions also show a degradation with a RMSE of 2.2 K at 1000 m but are slightly better than 1DVAR below 1300 m if AROME is used as background (fig. 13). Above 1300 m, an increase in the bias makes the regressions less accurate than both 1DVAR configurations. The best performance is found if radiosondes are used as backgrounds even though the RMSE values still reach 1.8 K at 1000 m.

In order to evaluate the added value brought by MWR to the background profile, figure 15 summarizes the performance of the

1DVAR retrievals either from RS or from AROME backgrounds compared to the AROME forecast errors. It also shows the performance of the "persistent" method where the last RS is used as an estimate of the current conditions. When the previously launched RS is used as background, the external weather station was added to the observation vector. As radiosondes were launched every 3 hours during the Passy-2015 campaign, it is interesting to investigate if MWR could still bring an information on atmospheric changes during the 3 hour time window. We note that even if radiosondes are launched every three hours,

significant changes in the boundary layer temperature profiles are observed between two adjacent RS with RMS values larger than 1 K below 1000 m and up to 3.6 K at 2 m in clear-sky conditions. Errors up to 1.6 K in RMSE in the temperature profile are also associated to the cloud-based inversion with the "persistent" method. 1D assimilation of MWR observations manages to significantly decrease the errors in the boundary layer mainly below 1500 m with values between 0.3 and 1 K in the first 1000 m. MWR observations can thus fill in the gap between 3 h radiosondes to provide valuable temperature profiles. This

result also demonstrates how the MWR temporal resolution is a necessity to complete our understanding and description of

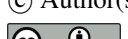



**Figure 12.** Vertical profiles of bias (left panel) and root-mean-square-errors (right panel) of 1DVAR retrievals using either AROME 1h forecast (blue line) or previously launched radiosonde (purple line) as background. 1DVAR retrievals are performed with bias correction and using all elevation angles (90 to 5.4 °). Comparison with linear regressions (black line). Results on 56 clear-sky temperature profiles.

the ABL diurnal cycle. As already shown, the AROME forecasts are significantly improved in clear-sky conditions but also in cloudy-sky. Even though errors up to 3 K are observed with the 1DVAR from the AROME forecasts, a significant improvement of the AROME forecasts is found in cloudy-sky conditions in the first 1200 m. Temperature errors are decreased from 3 K to 1 K in the boundary layer and 4.5 to 2.5 K at the cloud-based inversion. This result demonstrates the potential benefit of

5   assimilating MWR observations in NWP models both in clear-sky and cloudy-sky conditions.



**Figure 13.** Vertical profiles of bias (left panel) and root-mean-square-errors (right panel) of 1DVAR retrievals using either AROME 1h forecast (blue line) or previously launched radiosonde (purple line) as background. 1DVAR retrievals are performed with bias correction and using all elevation angles (90 to 5.4 °). Comparison with linear regressions (black line). Results on 25 cloudy-sky temperature profiles.

## 6.4 Examples of temperature profiles

In order to illustrate the capability of MWR to resolve deep near surface as well as elevated temperature inversions, figure 16 shows temperature profiles during two opposite weather regimes: convective and cloudy conditions the 07th of February at 06:04 UTC and stable clear conditions the 13th of February at 02:56 UTC. In each figure, temperature retrievals from different configurations (regression, 1DVAR from AROME forecast, 1DVAR from radiosonde) are compared to radiosonde. The *a priori* profile used in each configuration, either the previously launched radiosonde or the 1h AROME forecast is also shown. First of all, we can note the difficulty of ground-based MWR to resolve high level inversions. Neither the regression





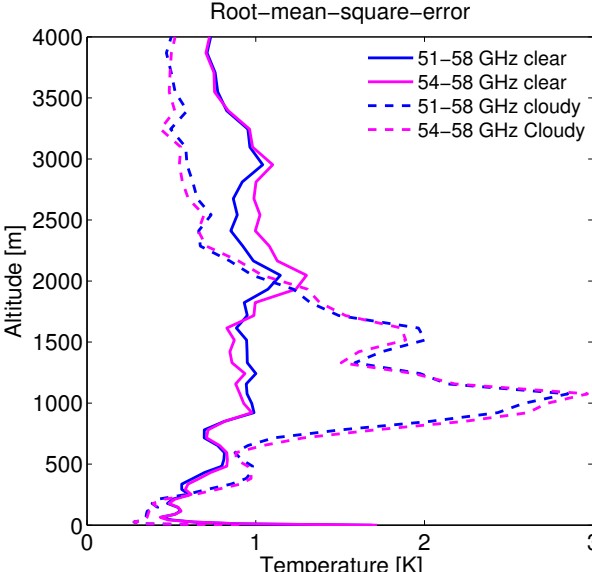

**Figure 14.** Vertical profiles of root-mean-square-errors of 1DVAR retrievals using AROME 1h forecasts as backgrounds. Either all V-band channels are used (blue) or only opaque channels (magenta). Results differentiated by clear-sky (solid lines) or cloudy-sky conditions (dashed lines)

or the AROME 1DVAR can catch the sharp inversions at 1000 m above ground level. Only the RS 1DVAR is able to catch it as the minimisation starts from a background profile already simulating an elevated inversion that is barely modified during the retrieval. Note that 1DVAR with AROME background shows an elevated inversion, though the AROME profile is almost linear. This already represents an improvement with respect to the AROME background which is too smooth. This limitation

5    is a well-known issue of MWR; Massaro et al. (2015) suggested the use of additional pressure and temperature observations from meteorological stations on the surrounding mountain slopes. An improvement can also be expected from more appropriate vertical correlations in the **B** matrix. In fact, correlations currently used probably smooth the increments and a reduction on the vertical correlation length should lead to a beneficial impact on the retrievals in such conditions. This approach will be investigated in the future.

10    Contrary to the high-level inversion, MWR can catch very well clear-sky deep near-surface temperature inversions as observed during the stable episode of the Passy-2015 campaign. Both 1DVAR and regressions capture well the structure of the profile even though 1DVAR retrievals are slightly more accurate than regressions. We can again note the significant improvement of the AROME profile in the lowest 500 m thanks to the MWR information content brought during the analysis.





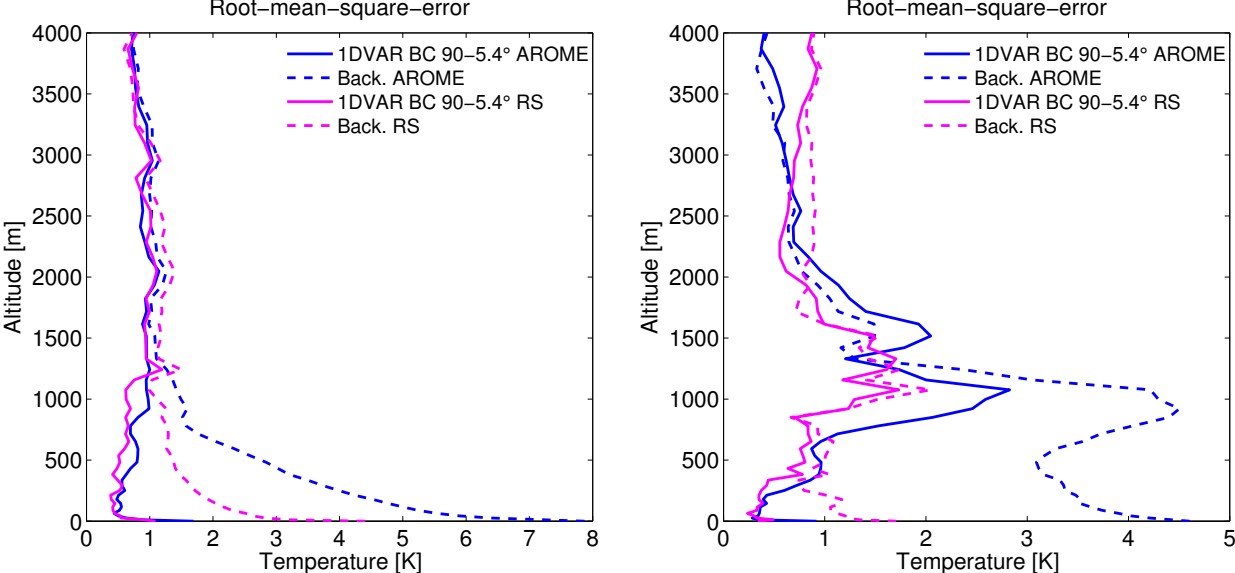

**Figure 15.** Vertical of profiles root-mean-square-errors of 1DVAR retrievals using the previously launched radiosonde (magenta line) or the AROME forecasts (blue line) as backgrounds. Comparison with the persistent method where the last RS is used as an estimate of the current conditions (dashed magenta) and the AROME forecast errors (dashed blue). Here, the weather station is included in the 1DVAR from RS to improve the temperature retrieval at the surface (1.5 m). Statistics on 56 clear-sky profiles (left panel) and 25 cloudy-sky profiles (right panel).

# 7 Conclusions

Within the Passy-2015 field campaign, a HATPRO ground-based microwave radiometer was operated in a deep Alpine valley constraining the measurement configuration due to surrounding mountains. A 1DVAR technique combining 1h forecasts of the convective scale model AROME and observations from the HATPRO MWR was tested and evaluated during two IOPs foccussing on wintertime stable boundary layers out of three months of instrumental deployment. An evaluation of the accuracy of the AROME model was first studied. A large underestimation of the surface cooling up to -12 K during the most stable episode was observed. This is a well-known issue of current NWP models that motivated, among other scientific questions, the preparation of the Passy-2015 campaign. This issue is currently investigated by the modelling community at CNRM and some significant leads for improvement have already been found. During the beginning of the IOP, AROME was found to smooth cloud-based inversions leading to larger errors at the cloud base around 1000 m while during clear-sky conditions the temperature inversion is not large enough. The measured brightness temperature (BT) measurements were compared with the ones simulated either from AROME 1h forecasts or RS and the ARTS radiative transfer model. The goal of this monitoring is to propose a bias correction to improve the retrieval of atmospheric profiles. The use of the AROME model to compute the instrumental bias correction was found inappropriate because the BT deviations for opaque channels are mainly driven



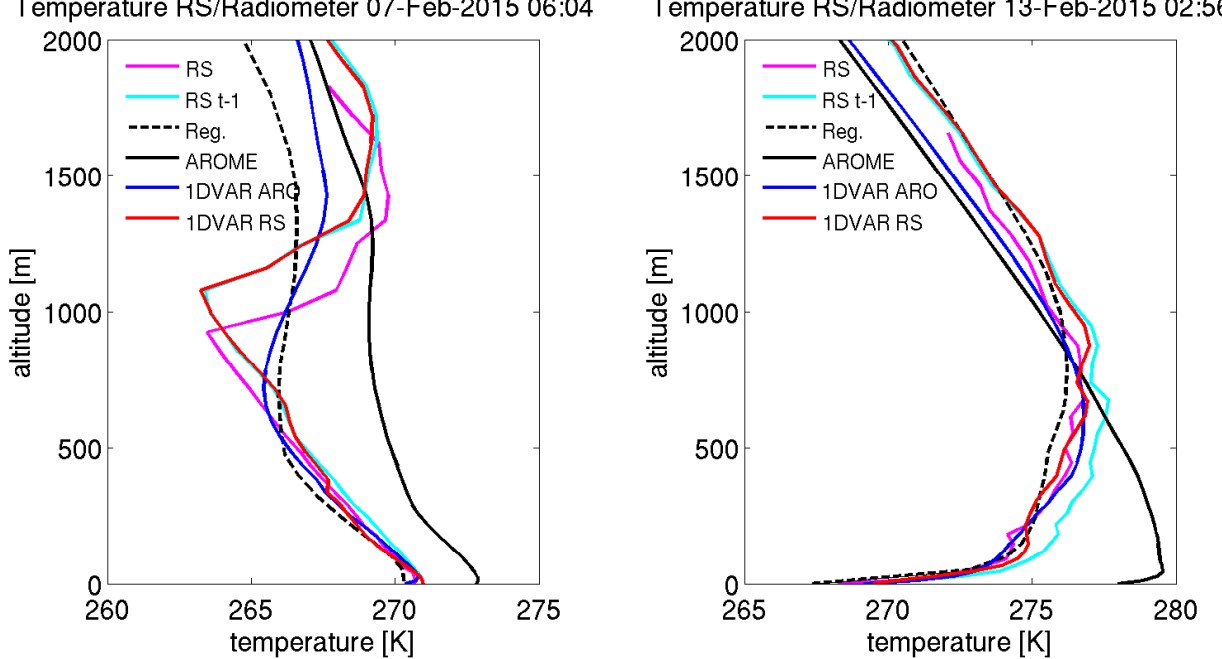

**Figure 16.** Vertical profiles of temperature during convective conditions (07 February 2015 at 06:04 UTC, top panel) or stable conditions (13 February at 02:56 UTC). Comparison between radiosonde profile (magenta), linear regression (dashed black line), 1DVAR from AROME forecasts (blue), 1DVAR from previous radiosonde (red). Background profiles corresponding to either the 1h AROME forecast (black) or the previously launched radiosonde (cyan) are also shown.

by the large forecast errors in the boundary layer during stable conditions. The instrumental bias was thus inferred from BT simulations with the RS launched during the campaign in clear-sky conditions. A large negative bias was observed for the most transparent channels with values up to -4.2 K (52.28 GHz, 90 °) while it is below 0.5 K for opaque channels and all elevation angles. Relatively low standard deviations (within 1 K) were observed for channels and elevation angles used in the retrieval

5 demonstrating that the biases can be safely removed by applying a constant bias correction. The bias is close to that found in previous studies. This demonstrates that the bias can be assumed constant as long as calibrations are performed properly. The second part of this study has evaluated 1DVAR retrievals in terms of bias and RMSE against collocated radiosondes. By exploiting the 1DVAR assimilation of MWR observations, the large forecast errors close to the surface (up to 8 K in RMSE) were decreased within 1 K through all the atmospheric profile except the surface temperature (1.6 K RMSE). This result is

10 really encouraging as it shows the high information content of MWR in the boundary layer specifically where the AROME forecasts are less accurate and could be improved by a dense network of ground-based instruments. 1DVAR retrievals were found to outperform linear regressions above 1000 m where RMSE values increase up to 2.5 K at 6000 m. Linear regressions show similar performance below 1000 m. The use of the elevation scanning mode was also found to significantly improve the retrievals below 1000 m while the use of a bias correction improves the retrievals below 2000 m. These last results are not





in agreement with the results in Martinet et al. (2015). However, the dataset used in this study contains mainly deep clear-sky near-surface temperature inversions for which low elevation angles can bring new information to the zenith mode. Finally, the use of an external weather station to constrain the temperature retrieval at the surface can decrease the RMSE values from 1.6 K to 1 K which includes the uncertainty due to relative distance.

In order to improve 1DVAR retrievals for processes study in the context of field campaigns, RS previously launched during the field campaign can be used as backgrounds in place of AROME forecasts. Starting from an *a priori* profile already closer to the true atmospheric state, a better estimation of the optimal atmospheric profile should be observed. In clear-sky conditions, this configuration leads to an improvement of 1DVAR retrievals below 1000 m with RMSE values below 0.6 K. An attempt of retrieving temperature profiles in cloudy-conditions was also studied. A significant degradation of both regressions and

1DVAR was found especially around 1000 m wih RMSE values around 2 K for regressions and 3 K for 1DVAR retrievals. This degradation is significantly reduced if RS are used as backgrounds. This degradation at 1000 m is probably due to cloud-based temperature inversions not caught by the MWR and does not seem to be directly related to large background errors in the liquid water absorption. This study confirms the known difficulty of MWR to capture elevated temperature inversions in cloudy-sky at the level of the valley crest (Crewell and Lohnert (2007), Massaro et al. (2015)) while highlighting the high capability of

MWR to catch clear-sky deep near-surface temperature inversions during stable boundary layers. MWR observations were also found to provide valuable information between two adjacent RS to catch significant changes in the ABL temperature profile. Regarding the scientific questions addressed in section 1, our results show that MWR are expected to bring valuable information into NWP models up to 3 km altitude but mainly in the first km both in clear-sky and cloudy-sky conditions. With an accuracy between 0.5 and 1 K in RMSE, our study has proved MWR to be capable of resolving deep near-surface temperature inversions

observed in complex terrain during stable boundary layer conditions. This accuracy can be obtained only if the MWR field of view is free of obstacles and is similar to what was observed in less complex terrain. Elevated temperature inversions are still challenging due to the decrease vertical resolution of the instrument with altitude. Using a more appropriate background already simulating an elevated inversion was already found to greatly improve the retrievals. In the future, extra work needs to be undertaken to decrease the correlation length of the background error covariance matrix which should improve the

retrievals. New generation of MWR also shows a larger sensitivity which is expected to help resolving elevated inversions. Finally, synergy with other passive and active instruments (infrared radiometers and lidars) is expected to improve the vertical resolution of the retrievals but only below clouds.

The results shown in this study are encouraging and demonstrate the potential for assimilating MWR in operational convective scale models even though studies on larger dataset and larger time periods should be investigated. The development of the

ground-based version of the fast radiative transfer model RTTOV (RTTOV-gb, De Angelis et al. (2016)) paves the way for future data assimilation of brightness temperature measurements which should bring more in the assimilation system than retrievals (Caumont et al. (2016)). In the context of urbanized valley, this study has proved the capability of MWR for long-term monitoring to improve our understanding of wintertime pollution events. Temperature gradients linked to the atmospheric stability could be used to better forecast wintertime pollution events.





In the future, 1DVAR retrievals will be extended to humidity and liquid water content. Improvement in the definition of the **R** and **B** matrices will also be carried out to be optimized with the weather regime.

## 8 Data availability

Data used in this paper are available on the Passy-2015 campaign website: http://passy.sedoo.fr.

5 *Acknowledgements.* We thank Alexander Haefele from MeteoSwiss for kindly providing us with the Payerne coefficient files used in the microwave radiometer retrievals. All the members of the CNRM/GMEI/LISA team are thanked for the maintenance and deployment of the MWR during field campaigns. The Passy-2015 field campaign was supported by ADEME through the French national programme LEFE/INSU and by METEO-FRANCE. We thank the cities of Passy and Sallanches for their kind support. The field experiment was led by CNRM while LEGI is the principal investigator of the LEFE/INSU project. Data are managed by SEDOO at Observatoire Midi-Pyrénées.
10 This work has also been supported through the COST Action ES1303 (TOPROF), supported by COST (European Cooperation in Science and Technology) and by SBFI-contract no. C15.0030.





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
