# Peer review of "Combining ground-based microwave radiometer and the AROME convective scale model through 1DVAR retrievals in complex terrain: an Alpine Valley case study"

_Atmospheric Measurement Techniques, 2017_

## Referee Comment (RC1) · Anonymous Referee #1 · 20 Jun 2017

Figure 16 - it is well known that the radiometer could resolve ground-based inversions but poor in depicting elevated temperature inversions. What would be the value added by 1DVAR?

One of the scientific objectives of the paper is to study the performance of radiometer in deep valley. However, the results look like similar observations have been reported for flat terrain as well. Which unique features of the measurements of the radiometer have been shown in the paper?

[Figure]

What new features have been found for retrieval at an elevation angle/boundary layer scans?

---

## Referee Comment (RC2) · Anonymous Referee #2 · 25 Jun 2017

Ground-based microwave radiometer (MWR) can perform continuous unattended measurements of the atmospheric temperature profile. The vertical resolution is best close to the ground and decreases with altitude. The manuscript demonstrates the quality of the temperature retrieval using a 1D-variational framework for the special conditions of the stable boundary layer in an Alpine valley. It makes uses of a rich radiosonde (RS) data set for evaluation purposes identifies systematic problems in the high-resolution weather forecast model Arome. Several investigations in respect to the sensitivity of the apriori (from Arome and RS) and for different conditions are performed which show

the potential of MWR to improve numerical weather forecast models.

The manuscript is well written, addresses an important topic and presents interesting results which can have high impact on the future observation network. In addition to some major points and recommendations I add a list of minor comments below.

1) In the introduction, the authors pose the question whether the surrounding mountains in the narrow valley affect the microwave observations. However, this is not really investigated in this study as only a comparison with radiosondes is made. Neither the atmospheric volume observed by the radiometer nor the flight track of the radiosonde is considered. One would need to calculate the effect of the antenna pattern or perform azimuth scans to see at which point the mountain slopes are in the field of view. In fact a simple calculation shows that even for transparent channels which receive radiation over the full extent of the atmosphere, mountains in 2 km wide valley should not provide a contribution to the main beam (2.5 deg FWHM). As side lobe suppression is -30 dB this is unlikely to contribute. More interesting is the guestion how strongly the true temperature field varies across the valley, e.g. is there any influence of mesoscale circulations or solar insolation? I do not expect the authors to perform an elaborated analysis in this respect but a more careful wording is necessary, e.g. "..thus can be safely deployed in complex terrain.."(p8, I5) or at p13,I1. It would be very interesting to know if the boundary layer scans performed in the two different directions as indicated in Figure 1 differ from each other?

2) As far as I know the HATPRO standard regression boundary layer temperature retrieval makes use of the inbuilt in-situ temperature measurements. This might explain the good performance despite the lack of bias correction. However, the HATPRO sensor should not be as accurate (representative) as the weather station - did you intercompare them?

3) The authors use radiosonde measurements down to 10 m, however, the first roughly 100 m of the radiosonde ascent suffer from the fact that the sensors are not fully vented,
Do the authors have some information on this from the tethered balloon measurements?

4) A short discussion on the vertical resolution of the MWR should be included. Löhnert and Maier (2012) smooth the radiosonde profile with the averaging kernels for comparison to take these effects into account but here you are interested in the optimal retrieval. In fact, this discussion would support your outlook that the inclusion of infrared radiometers and lidar could improve the vertical resolution (p26, I27), cf Barrera et al., AMT, 2016. I do not agree that this would help only below clouds: IR and lidar give information below clouds and thus the information content from the MWR could be exploited for the higher levels.

5) Fig. 8 shows a bias of about -5 K with STD of about 3 K. Looking at Fig.3 the difference between Arome and RS is certainly not Gaussian distributed - what is the impact on the B matrix? This might be discussed in respect to the question what the optimal way to build the B-matrix is, e.g. dependence on flow and diurnal cycle?

Minor comments:

p7,I28: "..and those extracted from" p8,I31: larger errors during unstable conditions". It is not necessary unstable but rather neutral with about 10 K difference between surface and about 1 km. You should rephrase and maybe just mention your stability index p10,I3: A more . p7,I19: One sentence explaining O-B would be helpful - maybe the last sentence of the paragraph? p10,I7: "Note, that the small amount.." p12,I4: Did you look at the variability of Arome within the valley? p12,I13: Did you check the retrieval at 0 m - this is a different site than in Martinet 2015? p13,I6: Which type of instrumental errors do you expect: drifts, calibration jumps? p14, I19: only for clear-sky? p15, I4: is observed -> is evident p15,I9-10: How is this done in detail - should be reproducible p15, I13: standard HATPRO linear regressions p15, I15: raw measurements makes me think of voltages...uncorrected TB + Tsurf (see point 2) p16, I1: I don't think plural of background exists. p16, I2: both IDVAR is confusing as three lines are shown. p16,

AMTD
117-24: This discussion describing Fig. 10 needs to be integrated with the previous paragraph (1-17) describing Fig. 9 as Fig.9 is only a blowup of Fig.10. This is just one story. For example, in line 6 the 0.7 K is mentioned that one can only see in Fig. 10 (actually it looks like 0.6). I also suggest to merge Fig. 9 and 10 to a 2x2 figure which would make it much easier. p16, I30-35: I am very surprised that there is a larger diurnal cycle in the difference between T2.5 and T1.5 than between T5 and T1.5? Did the values get mixed up by any chance? Please add the altitude of the HATPRO retrieval? p18, I10: At bit provocative: When you use the RS as a priori and also evaluate with an RS one could argue that systematic RS errors (time lag, calibration..) might be similar and therefore Arome has no chance? p24,I7: I do not understand how the valley constrained the measurement configuration - do you mean the difficulty to find a site with a free view?

Fig.6: This is much better than the first version but I cannot distinguish the two green (90 and 10.2) and the two red (42 and 19.2) lines - I think it would be better to reduce the number of angles and leave out the middle ones Fig.7: same color problem as for Fig. 6, better delete title (56) and add to caption Fig. 9: can't distinguish the two red lines Fig11-15 difficult to read -> larger

---

## Author Comment (AC1) · 19 Jul 2017

We would like to thank the reviewers for their time to review this manuscript and helpful suggestions to improve the manuscript. The reviewers questions are highlighted in bold, and the modifications to the manuscript in red. Figures 6 and 7 have been recomputed with a smaller number of elevation angles, and new colors have been chosen for figures 6, 7 and 9 to 13.

To answer to reviewer 2 comments, original figures 9 to 15 have been modified. We hope that the improvement we brought to the figures will now fit the reviewer comments to make the manuscript suitable for publication.

**Reviewer 1 comments :**

**Figure 16 - it is well known that the radiometer could resolve ground-based inversions but poor in depicting elevated temperature inversions. What would be the value added by 1DVAR?**

The main improvement from 1DVAR can be expected above 1 km altitude but for the two cases shown in the paper, the 1DVAR performs slightly better than regressions even below 1km. For the stable case (deep near surface temperature inversions), both 1DVAR and regressions can resolve the temperature inversions but the best accuracy is found with the 1DVAR particularly below 1 km altitude. For the cloud-based temperature inversion, the 1DVAR retrieval resolves better the inversion than the regressions. With both configurations (from the AROME background or the radiosonde), the inversion is more pronounced and closer to the radiosonde. Figures 9, 12 and 13 show that the value added by the 1DVAR is mainly above 1 km altitude where the RMSE stays within 1 K whereas it reaches 3 K with regressions.

**One of the scientific objectives of the paper is to study the performance of radiometer in deep valley. However, the results look like similar observations have been reported for flat terrain as well. Which unique features of the measurements of the radiometer have been shown in the paper?**

In such complex terrain we could have expected the measurements to be affected by surrounding mountains and one major result of this study is to show that MWR observations are not affected in such a narrow valley even going down to 5° elevation angles. In fact, previous papers deploying MWR in truly complex terrain are not abundant, from our knowledge only three : Kneifel et al 2010, Cimini et al 2011 and Massaro et al 2015. The study of Kneifel et al 2010 does not investigate the temperature profile retrievals and the radiometer is deployed at the mountain top above the crest.

In Cimini et al 2011, the terrain is more complex but the 1DVAR is investigated with a global NWP model at a 10 km horizontal resolution and only one elevation angle in addition to the zenith. The radiometer measurements do not go lower than 15° elevation angle which significantly limits the possible perturbance from surrounding mountains.

Massaro et al 2015 deploys the instrument in a valley with a free viewing angle up to 28 km whereas the Passy valley is only 5 to 6 km long in the Passy direction. This is also the first time, from our knowledge, that the instrument was operated scanning in two directions. In addition, Massaro et al 2015 only focussed on regressions without any comparison with the 1DVAR algorithm and the temperature gradients were smaller compared to what was observed during Passy. The study has thus shown that microwave radiometers are suitable for very complex terrain where mountains at less than 5 km from the instrument do not affect the quality of the measurements even if very low elevation angles are used (down to 5.4 °) , in particular during cold pool events.. This is also the first time that 1DVAR retrievals in a very complex terrain are evaluated using forecasts from a convective scale model.

**What new features have been found for retrieval at an elevation angle/boundary layer**

**scans?**

This study confirms that low elevation angles can be used in a complex terrain. The improvement brought by low elevation angles is equivalent to what was found in previous studies on flat terrain.

To summarize these comments, the following sentence has been added to the introduction :

**To that end, this is the first time that a MWR has been deployed in such a narrow valley (less than 5 km between the closest mountain slope and the instrument) with measurements going down to 5 ° elevation angle during which 1DVAR retrievals are performed from a convective scale model.**

---

## Author Comment (AC2) · 19 Jul 2017

We would like to thank the reviewers for their time to review this manuscript and helpful suggestions to improve the manuscript. The reviewers questions are highlighted in bold, and the modifications to the manuscript in red. Figures 6 and 7 have been recomputed with a smaller number of elevation angles, and new colors have been chosen for figures 6, 7 and 9 to 13.

To answer to reviewer 2 comments, original figures 9 to 15 have been modified. We hope that the improvement we brought to the figures will now fit the reviewer comments to make the manuscript suitable for publication.

**Reviewer 2 comments :**

**1) In the introduction, the authors pose the question whether the surrounding mountains in the narrow valley affect the microwave observations. However, this is not really investigated in this study as only a comparison with radiosondes is made. Neither the atmospheric volume observed by the radiometer nor the flight track of the radiosonde is considered. One would need to calculate the effect of the antenna pattern or perform azimuth scans to see at which point the mountain slopes are in the field of view. In fact a simple calculation shows that even for transparent channels which receive radiation over the full extent of the atmosphere, mountains in 2 km wide valley should not provide a contribution to the main beam (2.5 deg FWHM). As side lobe suppression is -30 dB this is unlikely to contribute. More interesting is the question how strongly the true temperature field varies across the valley, e.g. is there any influence of mesoscale circulations or solar insolation? I do not expect the authors to perform an elaborated analysis in this respect but a more careful wording is necessary, e.g. "..thus can be safely deployed in complex terrain.."(p8, l5) or at p13,l1. It would be very interesting to know if the boundary layer scans performed in the two different directions as indicated in Figure 1 differ from each other?**

We agree with the reviewer that the paper does not clearly investigate the effect of the mountain emission as azimuthal scans should have been performed. It is also true that mountain slopes should not contribute to the main beam at zenith. However, the mountain slopes were quite close in the Passy direction and it was interesting to investigate to what extent it impacts the MWR measurements. In fact, measurements at 54.94 GHz receives a small contribution of emission further than 6000 m and the first mountain slopes in the Passy direction are found at a distance of 4.5 km from the radiometer. As the sensitivity to the mountain slope is not detailed in the paper, we propose to remove « without being affected by surrounding mountain » in the introduction. The sentence «  MWR can thus be safely deployed in complex terrain and similar temperature  accuracy to that of flat and less complex terrain can be expected. » has been changed into : **MWR can thus be safely deployed in complex terrain and then similar temperature  accuracy to that of flat and less complex terrain can be expected, at least if the line of sight of the MWR is free of obstacles over distances larger than about 5 km.**

Figure 1 : Vertical profiles of bias (solid line) and root-mean-square-error (dashed line) of temperature profiles retrieved by regression against radiosondes. Profiles retrieved using all measurements (black) or only measurements made in the Passy direction or only measurements made in the Sallanches direction

[Figure]

The reviewer is right that it is interesting to investigate the differences in boundary layer scans. It was the reason for this alternance of observations in the valley.

Figure 1 shows that most differences between temperature profiles retrieved in each direction are located above the boundary layer. The agreement with radiosonde is degraded between 2 and 5 km with measurements in the Sallanches direction. Figure 2 also investigates the differences in brightness temperatures between Passy and Sallanches. We can see that very few differences are found for opaque channels. On the contrary, the lower the elevation angle is, the larger the BT differences are for transparent channels. The maximum of differences is found at 5.4° with BT measurements colder in the Sallanches direction than Passy. Figure 3 shows a time serie of BT measurements at 51.26 GHz and 5.4° of elevation and we can observe a time delay in the diurnal cycle between Sallanches and Passy, Sallanches getting warmer before Passy.

The brightness temperatures are also warmer in the direction of Passy all day long that can be explained by the local orography differences between Passy and Sallanches (for example the altitude of the valley bottom is lower at Sallanches than Passy and the valley is narrower at Sallanches) and maybe by the impact of the urbanization in the direction of Passy. Another source of explanation could be the formation of cold pool in Sallanches.

If we look at the time series of BT measurements at 58 GHz (figure 3, bottom panel), we observe the same diurnal cycle and measurements in both directions. The city of Passy at 2.5 km is too far to affect the measurement of opaque channels and the mountain emission is totally absorbed by previous atmospheric layers. We observe thus the diurnal cycle of the atmospheric temperature in the valley at less than 2 km of the radiometer. Even though temperature heterogeneities probably exist in the valley, it is likely that they are not strong enough over such a short distance to be captured by the MWR. This will be investigated in future work.

The differences observed at 5.4° should be investigated more in details in a future work to understand and explain how temperature heterogeneity in the valley can be linked to the atmospheric circulation. This perspective has been described in the discussion:

**Scanning in two different directions of the valley, MWR observations also offer the possibility of investigating temperature heterogeneity in the valley and how these differences are linked to the mesoscale circulation. This will be further investigated in a future study.**

Figure 2 : Bias (solid lines) and standard deviations (dashed lines) of BT observations between Sallanches and Passy (BT Sallanches – BT Passy)

[Figure]

Figure 3 : Time series of BT observations at 51.26 GHz (top panel) and 58 GHz (bottom panel) in the Sallanches direction (blue) or Passy direction (red) at angle 5.4°

[Figure]

[Figure]

**As far as I know the HATPRO standard regression boundary layer temperature rerieval makes use of the inbuilt in-situ temperature measurements. This might explain the good performance despite the lack of bias correction. However, the HATPRO sensor should not be as accurate (representative) as the weather station - did you intercompare them?**

This is right that the inbuilt in-situ temperature measurements can normally be included in the regressions. However, the HATPRO sensor is known to suffer from large errors due to an inappropriate ventilation of the sensor. Figure 4 shows the bias and standard deviation of the HATPRO in-situ sensor minus the 2.5 m tower measurements. We observe a large bias during the day with a maximum of 7 K around 12 UTC. The use of this in-built sensor would thus degrade the retrievals. For that reason, the manufacturer regressions are generally provided without the inclusion of this in-situ sensor and it would need a lot of extra work to properly recompute the regression coefficients including the surface temperature which could be applied by replacing the HATPRO in-situ station by an accurate weather station. However, we made some testing during our previous campaign and the inclusion of the temperature sensor only improves the result in the first 200m where regressions are already very accurate in our experiment.

Figure 4 : Bias and standard deviation of HATPRO in-built station minus tower measurements at 2.5 m.

[Figure]

**The authors use radiosonde measurements down to 10 m, however, the first roughly 100 m of the radiosonde ascent suffer from the fact that the sensors are not fully vented, Do the authors have some information on this from the tethered balloon measurements ?**

The Vaisala probe used in the radiosonde has a very short time response of less than 0.5 s and is naturally ventilated by the balloon ascent at a 5m/s speed. We are thus quite confident in the accuracy of the radiosonde measurements. We do not expect the measurements to be degraded in the first 100 m but only in the first 10 to 20 m.

The Vaisala probe used under the tethered balloon has a similar time response but the tethered balloon at about 1 m/s so it is even less ventilated than the radiosonde probe.

However the temperature profiles from RS and tethered balloon (both from Vaisala probes) have been compared from the ground to 45 m (maximum height of the tethered balloon when it was on the same site than the RS from 6/02 3pm to 10/02 noon). Maximum differences observed are lower than the sensors accuracy (0.5 deg C).

**A short discussion on the vertical resolution of the MWR should be included. Löhnert and Maier (2012) smooth the radiosonde profile with the averaging kernels for comparison to take these effects into account but here you are interested in the optimal retrieval. In fact, this discussion would support your outlook that the inclusion of infrared radiometers and lidar could improve the vertical resolution (p26, l27), cf Barrera et al., AMT, 2016. I do not agree that this would help only below clouds: IR and lidar give information below clouds and thus the information content from the MWR could be exploited for the higher levels.**

The reviewer is right and a small discussion about the limited vertical resolution of the MWR has been included. Firstly in section 3.3 : **It is important to note here that the retrieval grid is finer than the true instrumental resolution but matches the AROME model vertical resolution.**

and section 6.2 :
**Here the radiosonde profiles are interpolated to the retrieval grid without taking into account the smoothing errors due to the limited vertical resolution of the MWR. In fact, this resolution is approximately between 50 m and 500 m and only 4 independent pieces of information can be extracted from the signal {Lohnert et al 2012}. On the contrary, the temperature profile is sampled approximately every 10 m by the radiosonde. In the future, the averaging kernel matrix could be used to bring the radiosonde profile onto the MWR vertical resolution.**

**Fig. 8 shows a bias of about -5 K with STD of about 3 K. Looking at Fig.3 the difference between Arome and RS is certainly not Gaussian distributed - what is the impact on the B matrix? This might be discussed in respect to the question what the optimal way to build the B-matrix is, e.g. dependence on flow and diurnal cycle?**

Even though a bias is observed in the AROME forecast errors it does not necessarily imply that the forecast errors are not Gaussian. The only accurate way that is known nowadays to infer the forecast error distribution is to use an ensemble assimilation with an adequate number of members. It allows the evaluation of the Gaussianity by computing the distribution of the member differences with respect to the mean ensemble, all members being valid for the same forecast time. In our case, this ensemble assimilation is not available and we can only plot the frequency distribution of AROME minus RS differences mixing different days and forecast hours. This comparison is thus not-optimal for this estimation. However, we computed these differences for levels below 1000 m because the sample is too small above due to fewer radiosondes reaching high altitude (figure 6). We can see that the approximation of Gaussian distribution is respected for the lowest levels but a

negative skewness is observed at 1000 m. This is due to the small number of large errors at 1000 m at the beginning of the period corresponding to the cloud-based temperature inversions.

Figure 6 : Frequency distribution of AROME minus radiosonde differences at 500 m (left panel) and 1000 m (right panel) with corresponding Gaussian Distributions (red line).

[Figure]

Legrand et al 2016 recently evaluated this non-Gaussianity for the AROME model with a 90 member assimilation system. It was found that all control variables present some non-gaussianity but vorticity and divergence are more affected than temperature and humidity. This non-Gaussianity is reduced by the analysis process in areas constrained the most by observations. Even though non-gaussianity exists, current 3D-Var and 4D-Var system does not take into account this error. In addition, the B matrix should not be affected by the non-Gaussianity as it only affects the higher moments of the distribution (skewness and kurtosis) and not the ones used to compute the B matrix (mean and standard deviation).

The optimal way to build the B-matrix and to make it flow dependent and evolve with the diurnal cycle is again to use an ensemble assimilation to compute a new B matrix at each assimilation cycle (see Ménétrier et al 2014). This system has been extensively developed at Météo France in the last 5 years and should be operational by 2018. A B matrix flow-dependeng and varying in time will thus be available in the future and the impact on the 1D-Var will also be possible.

This problem has been discussed in the manuscript :
Section 6.1 :
**Non-Gaussianity can also affect forecast errors. Recently, Legrand et al 2016 evaluated the non-Gaussianity of analysis and forecast errors using a 90 member AROME ensemble assimilation. It was found that for all variables, non-Gaussianity exists but dynamical variables (vorticity and divergence) are more affected than temperature and humidity. The data assimilation reduces this non-Gaussianity at each cycle in regions well covered by observations. This ensemble assimilation does not exist for our period making complicated the evaluation of this Gaussianity in our context. However, it should affect higher moments of the error distribution than those used in the B matrix.**

Section 6.2 :

**The flow-dependency and diurnal cycle of forecast errors can be determined by implementing a real-time AROME ensemble assimilation system (Menetrier et al 2014). This is under-development and should be available next year.**

**Minor Comments**

All comments have been taken into account. The changes are highlighted in red in the manuscript.

**P10, l3 : A more**
Not clear to us what the referee refers to here; thus, we take no action
**p7, l19 : One sentence explaining O-B would be helpful.**
O-B monitoring has been introduced at the end of section 3.3 :
**Information about instrumental errors can be obtained by investigating differences between observations and simulations from background profiles (short-term forecasts or radiosondes). The monitoring of these differences called O-B (observations minus background) departures is essential to remove any systematic errors in the measurements, the forward operator or the background profiles (De Angelis et al 2017). They are investigated in section 5.**

**p12,l4: Did you look at the variability of Arome within the valley?**
The variability of the AROME profiles has been studied and was found very homogeneous justifying the use of only the closest AROME grid point in the valley. Figure 7 shows a time serie of the temperature values extracted at different levels for all AROME grid points within the Valley. Figure 8 shows a comparison of the different temperature profiles within the valley for two different days.

Figure 7 : Time series of AROME temperature values at three different levels (300 m top panel, 800 m middle panel, 1500 m bottom panel) for all AROME grid points within the valley (grey dots) and the AROME grid point closest to the microwave radiometer (black dots). This is compared to the radiosonde  (red) and the 2.5 m  tower (orange) measurements.

[Figure]

Figure 8 : Temperature profiles for all AROME grid points within the valley (grey lines) compared to the closest grid point to the microwave radiometer (black line) during two different days.

[Figure]

[Figure]

**P15, l19 : How is this done in detail, should be reproducible**

As the standard deviation of RS minus AROME differences are on top of the operational standard deviation above 2 km altitude, a new standard deviation vector is obtained by combining the operational standard deviation above 2 km and the one computed from AROME minus RS differences below 2 km altitude :

sigma_new = [sigma_oper(1:61), sigma_Passy(62:90)]

Covariances are then computed with the usual formulation :

cov(i,j)=cor(i,j) * sigma_new (i) * sigma_new (j)

with the operational correlations.

In order to make this more clear, we modified the sentence :
As the 1DVAR retrieval accuracy depends on how well the B matrix is defined, the diagonal terms of the B matrix (auto-covariance of the temperature errors) were simply replaced by the variance of the radiosonde minus AROME differences (i.e. the square of standard deviation values in Figure 8) below 2 km.

**Tower measurement differences between 2.5 and 5 m :**

The values were not mixed up but we are more confident with the accuracy of the 5 m measurements. They are based on a Socrima shield which is naturally ventilated whereas the 2.5 m sensor is a new development undertaken in our laboratory. It is composed of a PT100 probe and a thin wire in a prototype shield with a forced ventilation to retrieve temperature at a high temporal frequency. The experiment highlighted some problems that would request an improved shield

design. We suggest to modify the figure to remove the comparison with the measurements at 2.5 m and only compare with the measurements at 5 m. Figure 11 has been changed accordingly.

p18, l10 : **A bit provocative: When you use the RS as a priori and also evaluate with an RS one could argue that systematic RS errors (time lag, calibration..) might be similar and therefore Arome has no chance**

Of course using radiosonde measurements which show less background errors than AROME was expected to improve the retrievals especially at the cloud-based temperature inversions. However, the systematic RS errors should be negligible in this comparison. Here, we just want to illustrate how the 1DVAR can be applied in an experimental campaign adapting the background profile to obtain the best estimation of the true atmosphere and how to deal with elevated cloud-based inversion by using a more appropriate background. This has been highlighted in section 6.4 :

**Our study shows that another way of improvement is to use an external information to infer the presence of an elevated temperature inversion that will be incorporated in the background of the 1DVAR algorithm.**

**P24, l7 : I do not understand how the valley constrained the measurement configuration - do you mean the difficulty to find a site with a free view?**

By « constrained the measurement configuration » we mean a direction free of obstacles. We changed the sentence into :

**Within the Passy-2015 field campaign, a HATPRO ground-based microwave radiometer was operated in a deep Alpine valley making complex the instrumental deployment due to surrounding mountains.**

---

## Author Comment (AC3) · 19 Jul 2017

The comment was uploaded in the form of a supplement:
https://www.atmos-meas-tech-discuss.net/amt-2017-144/amt-2017-144-AC3-supplement.pdf

---

## Author Comment (AC4) · 19 Jul 2017

[revised manuscript text omitted]
, only three to our knowledge : Kneifel et al. (2010), Cimini et al. (2011) and Massaro et al. (2015). The study of Kneifel et al. (2010) does not investigate the temperature profile retrievals and the radiometer is deployed at the mountain top above the crest. In Cimini et al. (2011), the terrain is more complex but the 1DVAR is investigated with a global NWP model at a 10 km horizontal resolution and using only one elevation angle in addition to the zenith. The radiometer measurements do not go lower than 15 ° elevation angle which significantly limits the possible

[Figure]

**Figure 13.** Vertical of profiles root-mean-square-errors of 1DVAR retrievals using the previously launched radiosonde (red line) or the AROME forecasts (blue line) as backgrounds. Comparison with the persistent method where the last RS is used as an estimate of the current conditions (dashed red) and the AROME forecast errors (dashed blue). Here, the weather station is included in the 1DVAR from RS to improve the temperature retrieval at the surface (1.5 m). Statistics on 56 clear-sky profiles (left panel) and 25 cloudy-sky profiles (right panel).

perturbance from surrounding mountains. Massaro et al. (2015) deploys the instrument in a valley with a free viewing angle up to 28 km whereas the Passy valley is only 5 to 6 km long in the Passy direction and only focussed on regressions without any comparison with the 1DVAR algorithm. Temperature gradients were also smaller compared to those observed during Passy. This is also the first time, to our knowledge, that the instrument was operated scanning in two different directions.

[revised manuscript text omitted]

---

## Short Comment (SC1) · 27 Jul 2017

As I am still interested in ground-based MWR (also in complex terrain) I followed the discussion and just wanted to add a few comments:

During the COPS campaign we deployed a scanning HATPRO in the Murg valley (Black Forrest, Germany). This valley is of course not as deep as the Alpine valley in this study but still shows quite complex terrain and is quite narrow and we had to carefully deal with effects of the mountain slopes. We used these observations

to derive the variability of water vapor in the valley. The results of this work can be found in Kneifel et al., IEEE Geosci. Remote Sens. Lett., 6(1), 157-161, 2009 (http://ieeexplore.ieee.org/document/4717300/?reload=true) which might be worth to be added to the discussion.

The authors also mention for the work in Kneifel et al., 2010 that "the radiometer is deployed at the mountain top above the crest". This is not completely correct since the measurements were taken at the Environmental Research station Schneefernerhaus which is 300m below the mountain top. As one can see at the images on www.schneefernerhaus.de the mountain slopes are on one side very close to the instruments (less than appr. 30m away!). The authors are right that we didn't investigate the temperature profiles but both, the HATPRO and the dual polarization radiometer (DPR, 90+150 GHz) do regular elevation scans to one side down to low elevation angles where they could potentially hit the mountain crests at the other side. These elevation scans were explicitly used in several studies (e.g. Xie et al., JGR, 2012 to infer the orientation behaviour of snowflakes (http://onlinelibrary.wiley.com/doi/10.1029/2011JD016369/abstract;jsessionid=E5B64C021EB9E64AE4D8318E9571F1DE Considering the discussion of performing elevation scans in complex terrain (either for T-profile or scattering signatures of snowfall) the authors may consider also to include these references in their discussion.

Kind regards, Stefan Kneifel, University of Cologne

---

## Short Comment (SC2) · 28 Jul 2017

Dear Authors,

thank you very much for your reply. I think the question of how to use scanning MWR observations in a complex terrain is an important one and hence I think it is worth to discuss this issue thoroughly. Whether the MWR observations are then used to retrieve temperature, liquid water or water vapor does in my opinion not play a big role for this consideration.

You are right that we didn't look at the oxygen channels for T-profiles in the 2009 study but the overall problem is that one could hit some terrain at low elevation angles and this might contaminate your atmospheric signal. I don't think the steepnes of the valley really matters for that. In fact, the Murg valley was only 1km wide, so more narrow than the Alpine valley you measured in. When working with the K-Band channels for water vapor the problem is even bigger due to the wider beam width of the antenna at low frequencies. So even if at some elevation angle one should not hit the terrain, one still has to carefully consider the beam width and whether side lobes of the antenna could receive some surface emission from the terrain.

In Kneifel et al., 2009 we did not only use the 30° elevation observations as you write but also used elevation angles down to 5.4° in the north direction for the comparison with the aircraft measurements: "The HATPRO measurements from the lowest elevation angle could only be used in the north direction, because in the other directions, they were blocked by orography."

I mentioned the papers with data originating from the UFS because they confirm that high quality TB observations can be obtained in close proximity of complex terrain. The DPR and Hatpro were run in a synchronuous way which is the reason why we could use LWP from Hatpro in the scan direction to study its effect on snow scattering. The Hatpro at UFS also performes regular boundary layer scans including the lowest elevation angles for more than 10 years to derive T- and q-profiles but you are right that they weren't investigated in the studies I mentioned.

Kind regards, S. Kneifel

---

## Short Comment (SC3) · 28 Jul 2017

Dear Stefan Kneifel,

I agree with you that even if no low elevation angles are used you can have to deal with mountain slope possible perturbance in the measurements. In that sense we could have inferred that temperature profiles are expect not to be affected even with 5.4° elevation angle measurements.
Even though we could have concluded this just from the papers investigating IWV and snow, I think this is very indirect.

Thus I think it is still relevant to show these kind of studies specifically on temperature profiles.
And I do not totally agree that it does not matter if the valley is steep or not. In fact, if you are close to the mountain slope, with a not steep valley you can probably use lower elevation angles compared to a steep valley because in the steep valley you will be affected by the mountain even at 10 or 30° of elevation depending on the configuration.
This limits the places where you can deploy the instrument if you want good retrievals of temperature profiles with boundary layer scan. Not being able to use low elevation angles deteriorate the retrievals especially in the cases studied in the paper with very stable boundary layers. As we wanted to use all elevation angles, we could not deploy the instrument close to the city of Passy or Sallanches for example whereas it could have been more interesting than in the center of the valley.
Even though the papers you cite might show that we can expect no disturbance, I do not think from the results published we could have directly concluded that we can perform temperature profiles at the bottom of a valley using the lowest elevation angles.
From my point of view, the deployment the closest to our study is the one in Massaro et al 2015 and there is no mention to these UFS related papers.

I are sorry but I do not agree to say that I wrote that Kneifel et al 2009 only used 30° elevation angles. First of all nothing is mentioned in the manuscript with respect to the elevation angles except that the mountain elevation (which is not a big mountain) was smaller and only IWV was investigated which is true. Secondly in our answer to you I clearly mentioned that I noticed that elevation scans down to 5.4° degree are performed but the degradation in accuracy in the orography direction is not clearly investigated neither what accuracy in temperature we can expect from the free line of sight angles lower than 30° is mentioned. Thus, even though low elevation angles are performed, only based on the results of the paper, I am sorry but there is no proof that measurements below 30° of elevation angles are not affected by surrounding mountains and what accuracy I can expect in the temperature profiles in complex terrain.. There is no discussion and results about these low angles retrievals in the suggested papers.

Finally, showing that we can expect good temperature profiles at the bottom of a valley is only a tiny aspect of this paper as it deals with future data assimilation and modelling errors during stable conditions. Thus this papers is very different from the UFS studies except the complex terrain deployment and even in that sense both configurations are very different.

---

## Author Comment (AC5) · 28 Jul 2017

The comment was uploaded in the form of a supplement:
https://www.atmos-meas-tech-discuss.net/amt-2017-144/amt-2017-144-AC5-supplement.pdf

---

## Author Comment (AC6) · 28 Jul 2017

Dear Stefan Kneifel,

Thanks for your interest in the paper and relevant suggestions especially as an exhaustive state of the art is always difficult.

We modified the manuscript accordingly by citing both papers that you mention.

The main difference from Kneifel et al 2009 comes from the much more complex terrain encountered in Passy (2000 m difference of height between the valley and the moutain top and maximum 2 km wide) but also the fact that Kneifel et al 2009 only addresses integrated water vapor retrievals at an elevation angles above 30°. It would have been interesting to investigate the impact on the temperature profiles as elevation scans at 5.4° were discarded in the orography direction.

In Kneifel et al 2010 and Xie et al 2012, radiometers are deployed at 2650m above sea level. Even though the term « above the valley crest » was inappropriate it is interesting to highlight that we are exploring atmospheric conditions and instrumental deployment very different (at the bottom of a 2000 m steep sided valley).

In Xie et al 2012 elevation scans are also only performed above 15 ° with the DPR whereas nothing is mentioned for HATPRO (except if we missed this detail when reading the article). Thus, the retrieval of temperature profiles with low elevation angles down to 5° close to mountain slopes is not investigated neither.

In Kneifel et al 2010, again we could not find a mention to the elevation scans of the HATPRO instrument and temperature profiling was not investigated.

We suggest to modify the discussion into :

**Previous papers deploying MWR in complex terrain are not abundant, among them we can cite : Kneifel et al. (2009), Kneifel et al.(2010), Cimini et al. (2011), Xie et al. (2012) and Massaro et al. (2015). In Kneifel et al. (2009) the terrain is not as complex as in Passy with a maximum elevation of only 350 m and only integrated water vapor retrievals are investigated. Both studies of Kneifel et al. (2009) and Xie et al. (2012) do not investigate temperature profile retrievals neither and the radiometer is deployed at 2650 meters above sea level which differs from the deployment at the bottom of the 2000 m deep Passy valley. In Cimini et al. (2011), the terrain is more complex but the 1DVAR is investigated with a global NWP model at a 10 km horizontal resolution and using only one elevation angle in addition to the zenith. The radiometer measurements do not go lower than 15° elevation angle which significantly limits the possible perturbation from surrounding mountains. Massaro et al. (2015) deploys the instrument in a valley with a free viewing angle up to 28 km whereas the Passy valley is only 5 to 6 km long in the Passy direction and only focussed on regressions without any comparison with the 1DVAR algorithm. Temperature gradients were also smaller compared to those observed during Passy. This is also the first time, to our knowledge, that the instrument was operated scanning in two different directions down a steep sided valley.**

---

## Author Comment (AC7) · 7 Aug 2017

Regarding Dr. Stefan Kneifel additional discussion, our comments on the deployment of MWR in complex terrain have been moderated.

In the introduction we have changed the sentence :

To that end, this is the first time that a MWR has been deployed in such a narrow valley (less than 5 km between the closest mountain slope and the instrument) with measurements going down to 5° elevation angle during which 1DVAR retrievals are performed from a convective scale model.

Into :

**To that end, a MWR has been deployed in a narrow Alpine valley (less than 5 km between the closest mountain slope and the instrument) with measurements going down to 5° elevation angle. This is the first time 1DVAR retrievals are performed from a convective scale model in complex terrain during which large forecast errors are observed.**

In the conclusion we have changed :

In such complex terrain we could have expected the measurements to be affected by surrounding mountains and one **major** result of this study is to show that MWR observations are not affected in such a narrow valley even going down to 5° elevation angles

into :

In such complex terrain we could have expected the measurements to be affected by surrounding mountains and one **interesting** result of this study is to show that MWR observations are not affected in such a narrow valley even going down to 5° elevation angles

The discussion on the literature of complex terrain deployment has been changed from :

Previous papers deploying MWR in complex terrain are not abundant, only three to our knowledge : Kneifel et al. (2010), Cimini et al. (2011) and Massaro et al. (2015). The study of Kneifel et al. (2010) does not investigate the temperature profile retrievals and the radiometer is deployed at the mountain top above the crest. In Cimini et al. (2011), the terrain is more complex but the 1DVAR is investigated with a global NWP model at a 10 km horizontal resolution and using only one elevation angle in addition to the zenith. The radiometer measurements do not go lower than 15° elevation angle which significantly limits the possible perturbance from surrounding mountains. Massaro et al. (2015) deploys the instrument in a valley with a free viewing angle up to 28 km whereas the Passy valley is only 5 to 6 km long in the Passy direction and only focussed on regressions without any comparison with the 1DVAR algorithm. Temperature gradients were also smaller compared to those observed during Passy. This is also the first time, to our knowledge, that the instrument was operated scanning in two different directions.

Into :

Previous papers deploying MWR in complex terrain are not abundant, **among them we can cite** : Kneifel et al. (2009), Kneifel et al. (2010), Cimini et al. (2011), Xie et al. (2012) and Massaro et al. (2015). **In Kneifel et al. (2009) the terrain is not as complex as in Passy with a maximum elevation of only 350 m and only integrated water vapor retrievals are investigated. Both studies of Kneifel et al. (2009) and Xie et al. (2012) do not investigate temperature profile retrievals neither and the radiometer is deployed at 2650 meters above sea level which differs from the deployment at the bottom of the 2000 m deep Passy valley.** In Cimini et al. (2011), the

terrain is more complex but the 1DVAR is investigated with a global NWP model at a 10 km horizontal resolution and using only one elevation angle in addition to the zenith. The radiometer measurements do not go lower than 15° elevation angle which significantly limits the possible perturbation from surrounding mountains. Massaro et al. (2015) deploys the instrument in a valley with a free viewing angle up to 28 km and only focussed on regressions. **Regarding the Passy valley, the free line of sight is limited to 5 km in the Passy direction and 1DVAR retrievals from a convective scale model are performed. Temperature gradients were also larger compared to those observed in Massaro et al. (2015).**

---

## Author Comment (AC8) · 7 Aug 2017

The comment was uploaded in the form of a supplement:
https://www.atmos-meas-tech-discuss.net/amt-2017-144/amt-2017-144-AC8-supplement.pdf